# Mechanical Properties of Fiber-Reinforced Polymer (FRP) Composites at Elevated Temperatures

**Chuntao Zhang** [1,2,3,*] **, Yanyan Li** [1,2] **and Junjie Wu** [1,2]

1 Shock and Vibration of Engineering Materials and Structures Key Laboratory of Sichuan Province, Mianyang 621010, China
2 School of Civil Engineering and Architecture, Southwest University of Science and Technology, Mianyang 621010, China
3 Department of Mechanical Engineering, University of Houston, Houston, TX 77204, USA
* Correspondence: chuntaozhang@swust.edu.cn

**Abstract:** Many materials are gradually softened with increasing temperatures in the fire, which will cause severe damage. As a new fiber-reinforced polymer (FRP) composite, the change in mechanical properties of nanometer montmorillonite composite fiber-reinforced bars or plates at elevated temperatures has not been investigated. To obtain a more comprehensive study of the mechanical properties of FRP composites at high temperatures, experimental research on the nanometer montmorillonite composite fiber material under the tensile rate of 1 mm/min was conducted at target temperatures between 20 °C and 350 °C. Finally, the failure mode of the FRP composites after the tensile test was analyzed. The results demonstrate that the elevated temperatures had a major impact on the residual mechanical properties of fiber-reinforced polymer (FRP) composites when the exposed temperatures exceeded 200 °C. Below 200 °C, the maximum decrease and increase in the fracture load of fiber reinforced polymer (FRP) composites were between −34% and 153% of their initial fracture load. After exposing to temperatures above 200 °C, the surface color of fiber-reinforced polymer (FRP) composites changed from brown to black. When exposed to temperatures between 200 and 300 °C, the ultimate load of fiber-reinforced polymer (FRP) composites significantly increased from 731.01 N to 1650.97 N. Additionally, the stress−strain behavior can be accurately predicted by using the proposed Johnson−Cook constitutive model. The experimental results studied in this research can be applied to both further research and engineering applications when conducting a theoretical simulation of fiber-reinforced polymer (FRP) composites.

**Keywords:** fiber-reinforced polymer (FRP) composites; elevated temperatures; mechanical properties; reduction factor; constitutive model

## 1. Introduction

Generally speaking, the matrix resin of FRP typically consists of thermosetting resin and thermoplastic resin. In the fire environment, with the increase of temperature, the mechanical properties of composite materials mainly experience a three-time decrease. When the temperatures increase to the glass transition temperature $T_g$ of the resin matrix, it softens and enters the rubber state from the glass state. The ability of the resin matrix to transfer shear stress between reinforced fibers decreases, resulting in the first significant decrease in the mechanical properties of FRP. When the temperature is further increased to the resin decomposition temperature $T_d$ (about 300–400 °C), the resin matrix of FRP is gradually decomposed and carbonized and the toxic smoke is released, resulting in the second significant decrease in the mechanical properties of FRP. When the temperature continues to be increased, the resin matrix begins to burn and the combustion process releases more heat, resulting in the second significant decrease in the mechanical properties of FRP.

Studies on fiber-reinforced polymer (FRP) composites at high temperatures have been conducted. The bonding strength of the concrete matrix between carbon and glass fiber sheet changes after being exposed to temperatures of 20, 50, 65, and 80 °C, according to research by Leone et al. [1]. In addition, the test results demonstrated that the concrete matrix's transition temperature from shearing failure to cohesion failure was 65 °C. Salloum [2] conducted an axial compression test on FRP-strengthened cylinders (Φ: 100 cm; height: 200 mm) after exposing them to temperatures of 100 and 200 °C and were left at each target temperature for 1, 2, and 3 h separately. The test results indicated that external-bonded FRP materials' reinforcing efficacy was sensitive to high temperatures. At temperatures 2.5 times Tg, the ultimate capacity of concrete specimens enhanced with FRP was 25% less than at ambient temperatures. The above scholars mainly focused on the mechanical behavior of concrete elements reinforced with FRP. There are studies about the mechanical behavior of FRP as a standalone material at elevated temperatures. Pultruded carbon fiber-reinforced polymer (P-CFRP) specimens and CFRP tensile specimens manufactured with a hand lay-up method were subjected to a series of tests by Nguyen et al. [3,4] at temperatures that reached 700 °C. According to their findings, hand lay-up specimens' ultimate tensile strength and Young's modulus were reduced by 50% at 350 °C and 30% at 600 °C. Additionally, they demonstrated that the thermomechanical strength was lower than the residual strength for P-CFRP samples at the same degree of applied temperature. One of the pioneering studies regarding the behavior and characteristics of FRP materials at high temperatures that are utilized in industrial domains, such as the automotive, marine, and aerospace industries, was performed by Mouritz and Mathys [5]. At high temperatures, Shenghu and Zhishen [6] performed a series of tension tests on single-layer FRP sheets composed of GFRP, CFRP, and basalt-fiber reinforced polymer (BFRP). Among all the tested fiber-reinforced sheets, they concluded that the CFRP sheets had the highest strength and the GFRP sheets had the lowest strength [7]. At around 55 °C, all of the sheets' tensile strength significantly decreased, but no further substantial decline occurred as the temperature increased. The CFRP sheets had the highest residual strength, with almost 69% of their initial tensile strength. However, there still lacks the work of establishing the constitutive model to better predict the mechanical behavior of FRPs at elevated temperatures. In this research, we proposed a constitutive model based on the experimental results of FRPs at elevated temperatures to fill the research gap.

Currently, steel is a hot topic of high-temperature research. To more accurately evaluate the fire resistance of steel structures, a variety of experimental studies on the mechanical properties of different steels at high temperatures have been conducted [8–12]. After subjecting high-strength steels of S460, S690, and S960 to fire, Qiang et al. [13,14] performed tensile tests to investigate the residual mechanical properties after the fire. Test results demonstrated that the mechanical properties of the tested steels were affected by heating temperatures below 600 °C. In contrast to this research, Gunalan and Mahendran [15] demonstrated that the residual mechanical properties of high-strength steels decreased noticeably when the target temperature exceeded 300 °C. Chiew et al. [16] investigated high-strength S690 steel that suffered from the RQT process and found that the yield strength of S690 steel that suffered from the RQT process declined more slowly than that without the RQT procedure. Additionally, earlier research [17–26] has demonstrated that when exposure temperatures rose above a particular value, substantial changes in the residual mechanical properties of low-carbon steels and HSS were observed. According to previous studies [27–29], there is a critical temperature of various steels in the post-fire mechanical properties. When the exposure target temperatures do not exceed the critical temperature, the post-fire mechanical properties of various steels remain basically unaffected. However, when exposed to temperatures above the critical temperature, the post-fire mechanical properties change significantly, irrespective of the cooling methods.

In recent years, nanometer montmorillonite composite fiber materials have gradually been used in building structures. As a new fiber-reinforced polymer (FRP) composite, the change in mechanical properties of nanometer montmorillonite composite fiber-reinforced

bars or plates has not been investigated. Therefore, the mechanical behavior of nanometer montmorillonite composite fiber-reinforced plates subjected to different temperatures was studied. The experimental results provided in this paper can be applied to both further research and engineering applications when conducting theoretical analysis and numerical simulation of nanometer montmorillonite composite fiber-reinforced polymer (FRP) composites. In addition, this research is a part of the larger experimental program that aims to examine the mechanical characteristics and behavior of nanometer montmorillonite composite fiber-reinforced polymer (FRP) composites in various situations.

## 2. Test Program

### 2.1. Test Specimens

In this test, physical and mechanical properties of the nanometer montmorillonite composite fiber material, including the density $\rho$, Barcol hardness, fiber volume fraction, insoluble content of resin, water absorption, glass transition temperature $T_g$, tensile strength (main fiber direction) $f_{tm}$, tensile strength (secondary fiber direction) $f_{ts}$, compressive strength (main fiber direction) $f_{cm}$, compressive strength (secondary fiber direction) $f_{cs}$, and shock resistance are provided in Table 1. Furthermore, the decomposition temperature of the FRP-reinforced bonding colloid was less than 310 °C.

**Table 1.** Physical and mechanical properties of the nanometer montmorillonite composite fiber material.

| Performance | Performance Index |
|:---:|:---:|
| $\rho$ (kg·m$^{-3}$) | $\leq$2000 |
| Barcol hardness (HBa) | $\geq$50 |
| Fiber volume fraction (%) | $\geq$70 |
| Insoluble content of resin (%) | $\geq$90 |
| Water absorption (%) | $\leq$1.0 |
| $T_g$ (°C) | $\geq$290 |
| $f_{tm}$ (MPa) | $\geq$400 |
| $f_{ts}$ (MPa) | $\geq$10 |
| $f_{cm}$ (MPa | $\geq$100 |
| $f_{cs}$ (MPa) | $\geq$15 |
| Shock resistance (kJ·m$^{-2}$) | $\geq$240 |

To better comprehend how high temperatures affect composites made of FRP, experiments were conducted. The size and the form of the FRP specimens are shown in Figure 1, and the specimens were fabricated in a thickness of 5 mm, which followed the specifications outlined in GB/T 228.1-2010 [30]. The experiments included 24 specimens and 8 target temperatures. Three specimens were loaded at each temperature to reduce the test error.

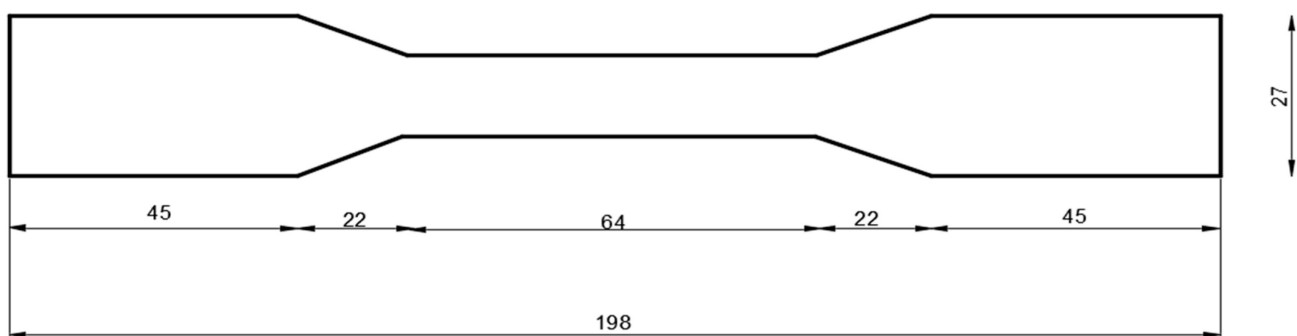

**Figure 1.** The FRP tensile coupon specimen (mm).

### 2.2. Test Details

To simulate different fire accidents, the ambient temperature and seven target temperatures—50 °C, 100 °C, 150 °C, 200 °C, 250 °C, 300 °C, and 350 °C—were con-

sidered herein. During high-temperature tensile tests, the specimen was first heated to a predetermined temperature at a heating rate of 15 °C/min. To ensure uniform temperature over the entire gauge length, the specimen was kept for about 30 min at each target temperature, and the elevated temperatures remained unchanged based on the thermocouples in the test equipment. Then, the specimen was loaded until it failed, during which the target temperatures were unchanged since the specimen was still in the test equipment. Both the displacement and engineering strain of the FRPs were output by the computer. As shown in Figure 2, this experiment was conducted by an ETM series electronic universal testing machine, and the displacement control method was used to test the specimens at a constant rate of 1 mm/min until fracture, which conformed to the requirements of GB/T 228.1-2010 [30]. Both the displacement and engineering strain of the FRPs were output by the computer. Based on the experimental results, the mechanical properties of fiber-reinforced polymer (FRP) composites were discussed.

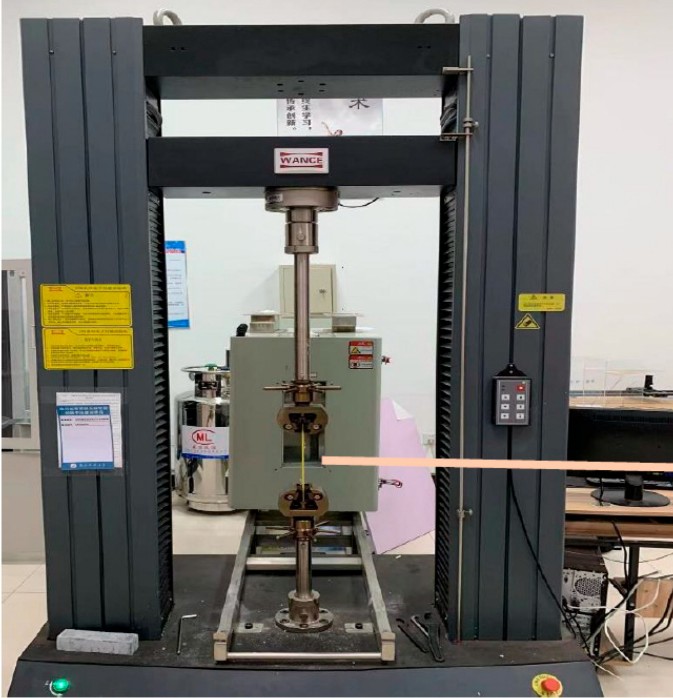 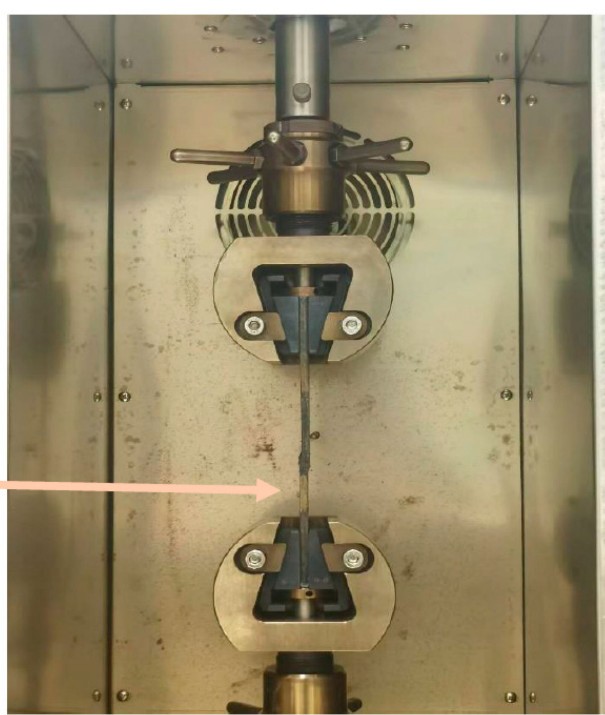

**Figure 2.** Test machine.

### 3. Experimental Results

*3.1. Load−Displacement Curves*

The tensile load−displacement curves of FRP composites at elevated temperatures of 50–350 °C were obtained and compared with the as-received state, as illustrated in Figure 3. When exposed to temperatures below 200 °C, the tensile load−displacement curves of fiber-reinforced polymer (FRP) composites exhibited minor differences, and the size and shape of tensile load−displacement curves were similar compared with the initial state, as shown in Figure 3a–d. In contrast, when exposed to temperatures of 300 °C, the tensile load−displacement curves of FRP composites exhibited significant differences when compared to the FRP composites in their as-received state, as illustrated in Figure 3f. Moreover, when exposed to temperatures of 250 °C, the tensile load of fiber-reinforced polymer (FRP) composites slightly increased when compared to that of the FRP composites at ambient temperature with the increased displacement, as illustrated in Figure 3e. However, the tensile load of FRP composites decreased sharply with the increased displacement when exposed to temperatures of 350 °C, which indicated that the tensional strength and ductility increased significantly at those target temperatures, as shown in Figure 3e,f. The reason for this phenomenon is that the matrix bonded by

the fiber resin changed with the increase of temperature and the glue or epoxy resin was softened at high temperatures. Therefore, the critical temperature of FRP composites was 200 °C, and the ultimate bearing temperature of FRP composites was 300 °C.

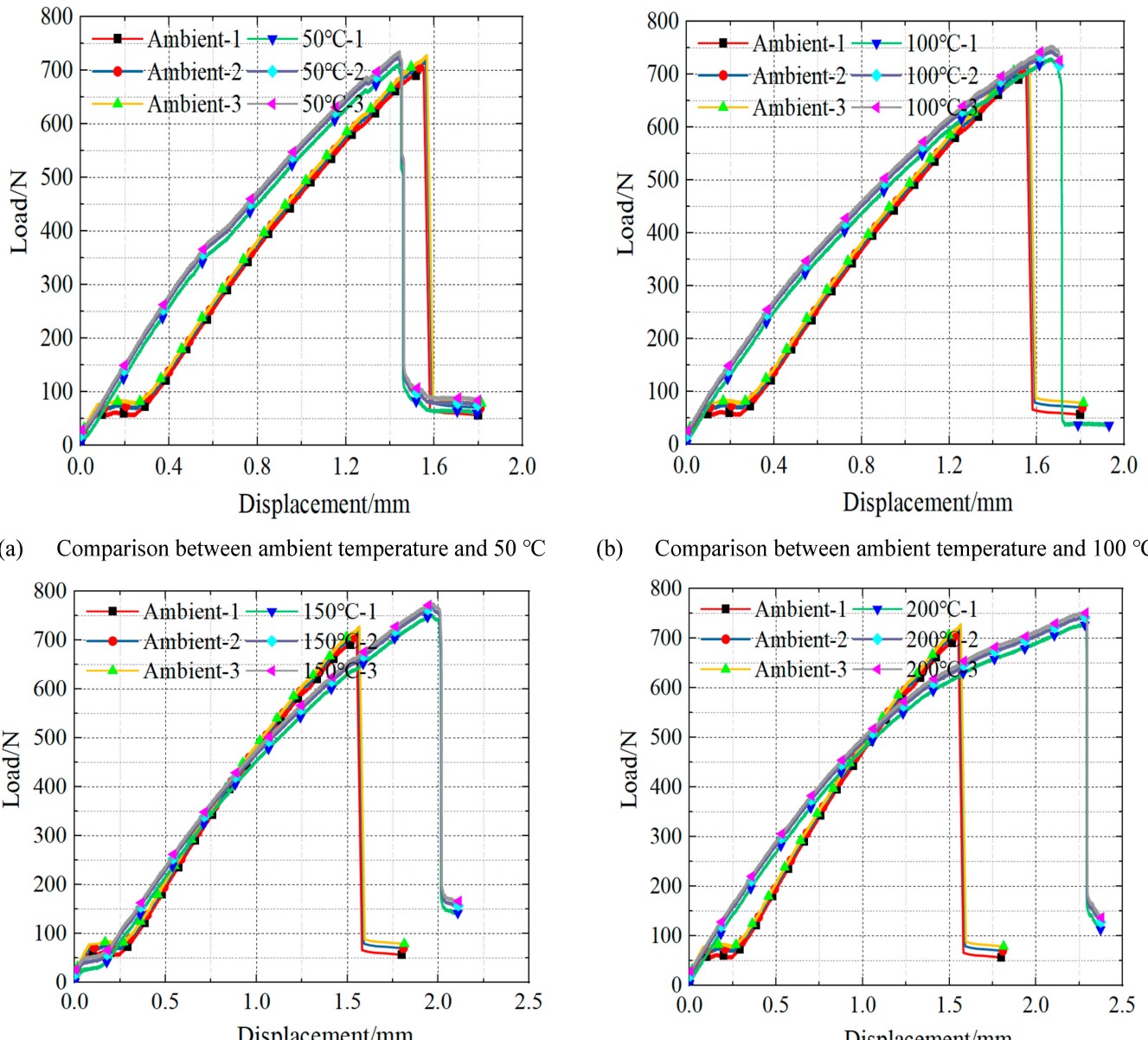

(a)   Comparison between ambient temperature and 50 °C

(b)   Comparison between ambient temperature and 100 °C

(c)   Comparison between ambient temperature and 150 °C

(d)   Comparison between ambient temperature and 200 °C

**Figure 3.** *Cont.*

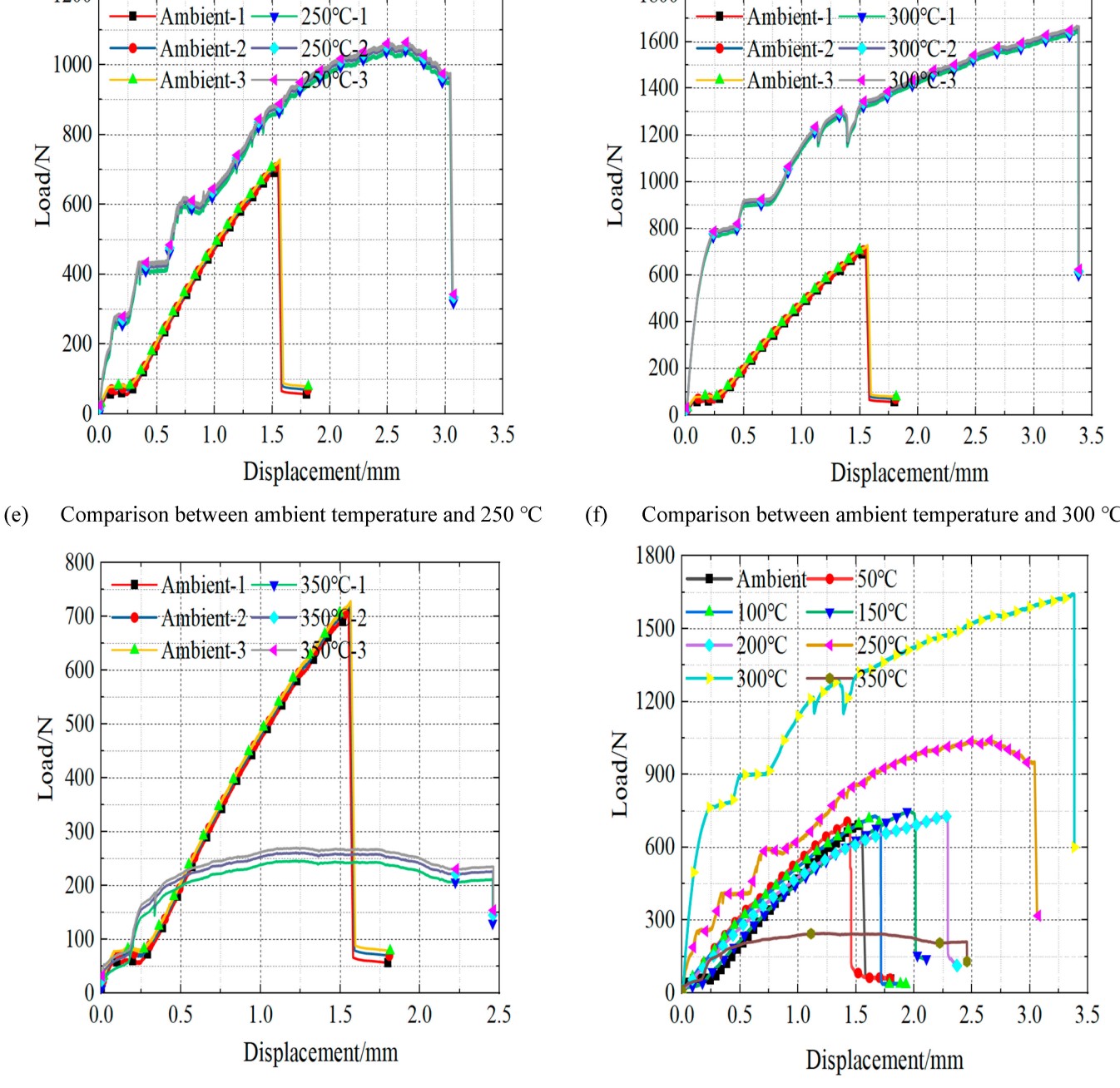

(e)     Comparison between ambient temperature and 250 °C

(f)     Comparison between ambient temperature and 300 °C

(g)     Comparison between ambient temperature and 350 °C

(h)     Comparison summary chart of all the target temperatures

**Figure 3.** Load−displacement curves.

### 3.2. Visual Observations

Figure 4 exhibits the visual observations of fractured specimens at different elevated temperatures. The surface color of the FRP composites was significantly affected by the elevated temperatures. The surface color of FRP composites at the ambient temperature was fully brown, and it gradually changed to black when exposed to elevated temperatures between 50 °C and 200 °C. After exposure to temperatures above 200 °C, the surface color of fiber-reinforced polymer (FRP) composites changed to fully black. It is worth noting that the fibers on the surface of the fiber composite material were shed after exposure to temperatures above 200 °C. Moreover, with the increase in temperature, the phenomenon of spalling at the center fracture position of fiber-reinforced polymer (FRP) composites was more obvious.

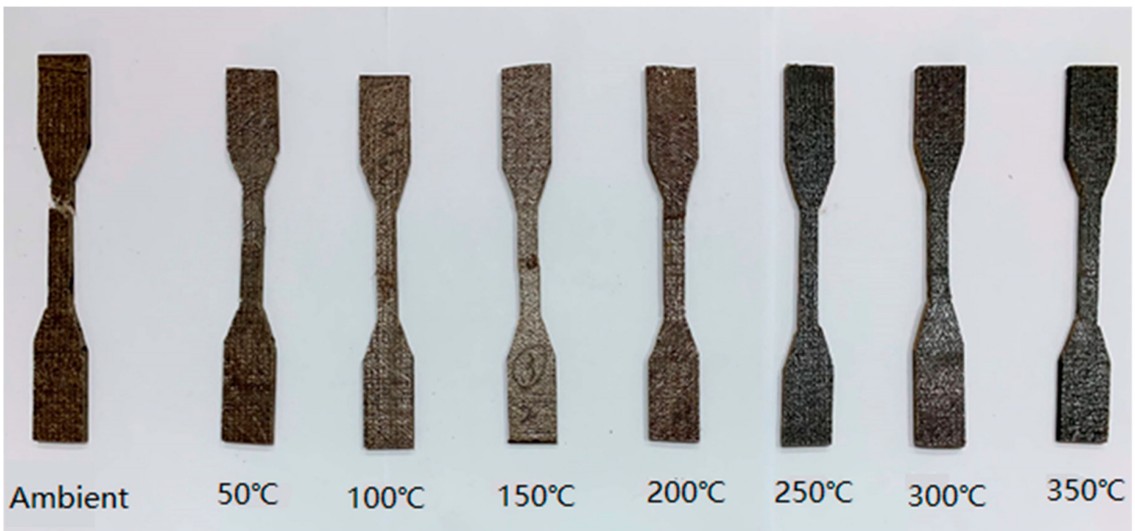

**Figure 4.** Specimens after experiencing elevated temperatures and tensile loading.

## 4. Discussion of Results

### 4.1. Ultimate Load

The maximum load that a structure or component can sustain is referred to as the ultimate load, and the component will enter an unstable state if the maximum load is reached. The ultimate load and residual factors of FRP composites at high temperatures are shown in Table 2. The variations in the reduction factor of the ultimate load are illustrated in Figure 5a.

The ultimate load of the specimens remained basically unchanged when exposed to temperatures below 200 °C, and the variation in the reduction factors of ultimate load did not exceed 6% when compared to the initial ultimate load of the FRP composites. However, when exposed to temperatures between 200 and 300 °C, the ultimate load of the FRP composites significantly increased from 731.01 N to 1650.97 N and increased by 133.51% of the initial ultimate load, which indicates that the FRP composites experienced a strengthening process. The reason for this phenomenon is that the matrix bonded by the fiber resin changed with the increase in temperature, when exposed to temperatures below 200 °C. The mechanical properties slightly increased, especially when the exposure temperatures are between 200 and 300 °C. This could be attributed to that the bonding effects of the nanometer montmorillonite and the fiber material were most obvious, which led to the increment of ultimate load. Notably, the ultimate load of FRP composites significantly decreased from 1650.97 N to 252.24 N and reduced 64.32% of the initial ultimate load when exposed to the temperature of 350 °C. This could be attributed to the fact that the bonding of the nanometer montmorillonite and the fiber material was softened at this temperature and the resin matrix entered the rubber state from the glass state, in which the transition temperature Tg was nearly 300 °C based on the test results.

**Table 2.** Ultimate loads and residual factor sof FRP composites at elevated temperatures.

| Temperature (°C) | Ultimate Load (N) | | | | Residual Factor ($F_{u,T}/F_{u,20}$) | | | |
|---|---|---|---|---|---|---|---|---|
| | Group-1 | Group-2 | Group-3 | Average | Group-1 | Group-2 | Group-3 | Average |
| 20 | 706.87 | 710.24 | 703.92 | 707.01 | 1.00 | 1.00 | 1.00 | 1.00 |
| 50 | 709.99 | 708.34 | 705.67 | 708.00 | 1.00 | 1.00 | 1.00 | 1.00 |
| 100 | 727.13 | 720.61 | 725.85 | 724.53 | 1.03 | 1.02 | 1.03 | 1.02 |
| 150 | 752.58 | 750.37 | 755.60 | 752.85 | 1.06 | 1.06 | 1.07 | 1.06 |
| 200 | 730.25 | 728.26 | 734.52 | 731.01 | 1.03 | 1.03 | 1.03 | 1.03 |
| 250 | 1036.51 | 1031.27 | 1049.58 | 1039.12 | 1.47 | 1.46 | 1.48 | 1.47 |
| 300 | 1643.84 | 1658.31 | 1650.77 | 1650.97 | 2.33 | 2.35 | 2.33 | 2.34 |
| 350 | 244.75 | 253.67 | 258.31 | 252.24 | 0.35 | 0.36 | 0.37 | 0.36 |

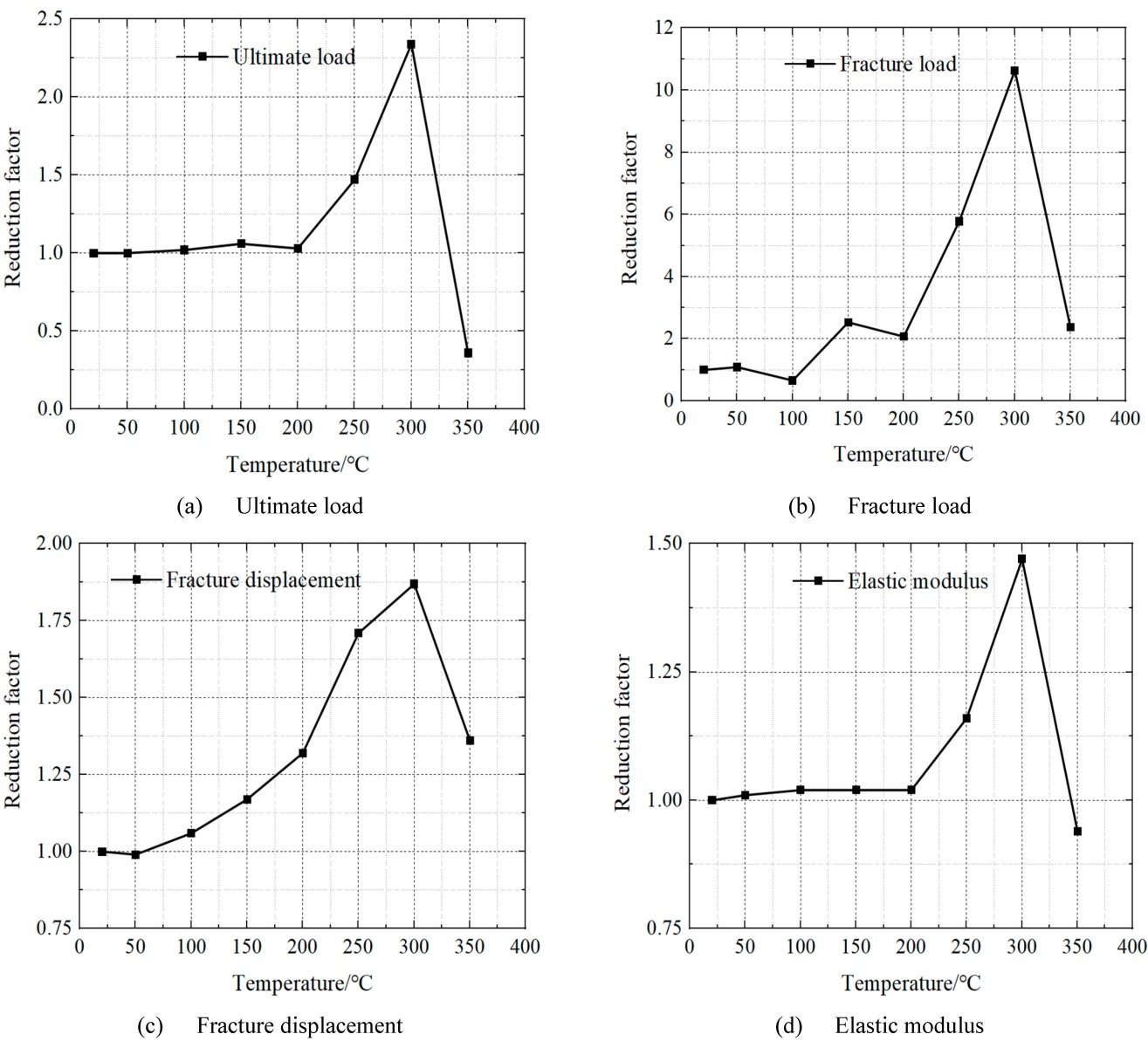

**Figure 5.** Residual mechanical properties of FRP composites at elevated temperatures.

### 4.2. Fracture Load

The critical load at which a material fails when subjected to continuous loading is referred to as the fracture load. The fracture load and the residual factor of FRP composites at high temperatures are listed in Table 3. The residual fracture load factors are plotted in Figure 5b.

When exposed to temperatures below 200 °C, the fracture load of fiber-reinforced polymer (FRP) composites remained basically the same as the initial state. The maximum decrease and increase in the fracture load of the FRP composites were between −34% and 153% of their initial fracture load, as depicted in Table 3. However, the maximum fracture load of the FRP composites was 541.35 N and was 963.26% of their initial fracture load when exposed to temperatures between 200 and 300 °C, which is consistent with the phenomenon of ultimate load. This could be attributed to the fact that the bonding effects of the nanometer montmorillonite and the fiber material were most obvious at those temperatures, which led to the increment of fracture load. Furthermore, the fracture load of the FRP composites significantly decreased from 597.55 N to 133.62 N when exposed to temperatures of 350 °C. This could be attributed to the fact that the bonding of the

nanometer montmorillonite and the fiber material was softened at this temperature and the resin matrix entered the rubber state from the glass state, in which the transition temperature Tg was nearly 300 °C based on the test results.

**Table 3.** Fracture loads and residual factors of the FRP composites at elevated temperatures.

| Temperature (°C) | Fracture Load (N) | | | | Residual Factor ($Ff_{,T}/Ff_{,20}$) | | | |
|---|---|---|---|---|---|---|---|---|
| | Group-1 | Group-2 | Group-3 | Average | Group-1 | Group-2 | Group-3 | Average |
| 20 | 56.88 | 58.51 | 53.22 | 56.20 | 1.01 | 1.04 | 0.95 | 1.00 |
| 50 | 60.25 | 61.38 | 62.51 | 61.38 | 1.07 | 1.09 | 1.11 | 1.09 |
| 100 | 36.36 | 38.52 | 36.33 | 37.07 | 0.65 | 0.69 | 0.65 | 0.66 |
| 150 | 142.31 | 139.82 | 144.33 | 142.15 | 2.53 | 2.49 | 2.57 | 2.53 |
| 200 | 113.23 | 118.33 | 120.87 | 117.48 | 2.01 | 2.11 | 2.09 | 2.07 |
| 250 | 319.33 | 325.41 | 330.57 | 325.10 | 5.68 | 5.79 | 5.88 | 5.78 |
| 300 | 600.87 | 581.34 | 610.45 | 597.55 | 10.69 | 10.34 | 10.86 | 10.63 |
| 350 | 130.47 | 133.91 | 136.47 | 133.62 | 2.32 | 2.38 | 2.43 | 2.38 |

### 4.3. Fracture Displacement

The fracture displacement is the displacement that corresponds to the fracture load. Table 4 lists the fracture displacement and the residual factor of the FRP composites at high temperatures. The residual fracture displacement factors are plotted in Figure 5c.

**Table 4.** Fracture displacements and residual factors of the FRP composites at elevated temperatures.

| Temperature (°C) | Fracture Displacement (mm) | | | | Residual Factor ($X_{f,T}/X_{f,20}$) | | | |
|---|---|---|---|---|---|---|---|---|
| | Group-1 | Group-2 | Group-3 | Average | Group-1 | Group-2 | Group-3 | Average |
| 20 | 1.80 | 1.81 | 1.81 | 1.81 | 1.00 | 1.00 | 1.00 | 1.00 |
| 50 | 1.79 | 1.80 | 1.79 | 1.79 | 0.99 | 1.00 | 0.99 | 0.99 |
| 100 | 1.93 | 1.92 | 1.91 | 1.92 | 1.07 | 1.06 | 1.06 | 1.06 |
| 150 | 2.11 | 2.11 | 2.12 | 2.11 | 1.17 | 1.17 | 1.17 | 1.17 |
| 200 | 2.37 | 2.38 | 2.40 | 2.38 | 1.31 | 1.32 | 1.32 | 1.32 |
| 250 | 3.07 | 3.10 | 3.12 | 3.10 | 1.70 | 1.72 | 1.73 | 1.71 |
| 300 | 3.38 | 3.37 | 3.36 | 3.37 | 1.87 | 1.87 | 1.86 | 1.87 |
| 350 | 2.46 | 2.44 | 2.47 | 2.46 | 1.36 | 1.35 | 1.37 | 1.36 |

Contrary to what was discussed above, the fracture displacement and the residual factor of the FRP composites at high temperatures gradually increased with the increasing temperature. Particularly when the FRP composites were exposed to 300 °C, the maximum increase in fracture displacement was 1.56 mm and 87% of their initial fracture displacement. This could be attributed to the fact that the bonding effects of the nanometer montmorillonite and the fiber material were most obvious at those temperatures, which led to the mass increment of ductility, causing the fracture displacement to increase. It is worth noting that when exposed to temperatures above 300 °C, the fracture displacement of the FRP composites decreased from 3.37 mm to 2.46 mm. This could be attributed to the fact that the bonding of the nanometer montmorillonite and the fiber material was softened at this temperature and the resin matrix entered the rubber state from the glass state, in which the transition temperature Tg was nearly 300 °C based on the test results.

### 4.4. Elastic Modulus

The elastic modulus is termed as the ratio of engineering stress to engineering strain in the elastic deformation stage during the tensile process. The elastic modulus and residual factors of the FRP composites at high temperatures are listed in Table 5. The residual elastic modulus factors are depicted in Figure 5d.

**Table 5.** Elastic moduli and residual factors of FRP composites at elevated temperatures.

| Temperature (°C) | Elastic Modulus (MPa) | | | | Residual Factor ($E_T/E_{20}$) | | | |
|---|---|---|---|---|---|---|---|---|
| | Group-1 | Group-2 | Group-3 | Average | Group-1 | Group-2 | Group-3 | Average |
| 20 | 799.21 | 802.34 | 803.11 | 801.55 | 1.00 | 1.00 | 1.00 | 1.00 |
| 50 | 810.35 | 815.24 | 808.65 | 811.41 | 1.01 | 1.02 | 1.01 | 1.01 |
| 100 | 812.77 | 817.65 | 815.37 | 815.26 | 1.01 | 1.02 | 1.02 | 1.02 |
| 150 | 813.38 | 815.48 | 816.74 | 815.20 | 1.01 | 1.02 | 1.02 | 1.02 |
| 200 | 820.39 | 819.35 | 821.22 | 820.32 | 1.02 | 1.02 | 1.02 | 1.02 |
| 250 | 925.33 | 925.49 | 929.64 | 926.82 | 1.15 | 1.15 | 1.16 | 1.16 |
| 300 | 1173.65 | 1182.37 | 1188.29 | 1181.44 | 1.46 | 1.48 | 1.48 | 1.47 |
| 350 | 750.24 | 758.41 | 749.59 | 752.75 | 0.94 | 0.95 | 0.94 | 0.94 |

The elastic modulus of the FRP composite specimens remained basically unchanged when exposed to temperatures below 200 °C, and the variation in the residual factors of elastic modulus did not exceed 2% when compared to the initial ultimate load of FRP composites. However, when exposed to temperatures between 200 and 300 °C, the elastic modulus of the FRP composites significantly increased from 820.32 MPa to 1181.44 MPa and increased by 47.39% of the initial elastic modulus, which indicated that the FRP composites experienced a strengthening process. This could be attributed to the fact that the bonding effects of the nanometer montmorillonite and the fiber material were most obvious at those temperatures, which led to the increment of elastic modulus. Notably, the elastic modulus of FRP composites significantly decreased from 1181.44 MPa to 752.75 MPa and reduced by 6.1% of the initial elastic modulus when exposed to the temperature of 350 °C. This could be attributed to the fact that the bonding of the nanometer montmorillonite and the fiber material was softened at this temperature and the resin matrix entered the rubber state from the glass state, in which the transition temperature Tg was nearly 300 °C based on the test results.

## 5. Constitutive Modeling

### 5.1. Johnson−Cook Model

Johnson and Cook initially proposed the Johnson−Cook model in 1983 [31,32]. The various stress−strain relationships of metallic materials in situations of large deformation, high strain rates, and elevated temperature could be properly described by this model. It has been frequently utilized, since it was first introduced due to its simple form. This constitutive model was expressed as follows:

$$\sigma\left(\varepsilon^p, \dot{\varepsilon}, T\right) = \left[A + B(\varepsilon^p)^n\right]\left[1 + C\ln\left(\frac{\dot{\varepsilon}}{\dot{\varepsilon}_R}\right)\right]\left[1 - \left(\frac{T - T_R}{T_m - T_R}\right)^m\right] \tag{1}$$

where $n$ is the constant coefficient of strain hardening, $C$ is strain rate strengthening coefficient, $m$ is thermal softening coefficient, $A$ is the nominal yield stress (MPa) in the tensile process, $B$ is the strain hardening constant (MPa), and $\sigma$ and $\varepsilon$ are the engineering stress and plastic strain, respectively, $\dot{\varepsilon}_R$ and $T_R$ are the reference strain rate and reference deformation temperature, respectively, $T_m$ is the melting temperature of the various metallic materials, and $T$ is the experimental temperature in the test. The three terms in the constitutive model, read from left to right, represent the effects of heating of elevated temperatures, strengthening of strain rate, and strain hardening of flow stress [33,34]. In this study, the reference temperature and strain rate in this experiment were $T_R$ = 293 K and $\dot{\varepsilon}_R$ = 0.005 s$^{-1}$, respectively. Under this experimental circumstance, $A$ = 9.43 MPa and $T_m$ = 1300 K.

### 5.1.1. Identification of Parameters B and n

Under the deformation rate and temperature $\dot{\varepsilon} = \dot{\varepsilon}_R = 0.005\ \mathrm{s}^{-1}$, $T = T_R = 293$ K. Equation (1) was transformed to the following:

$$B\varepsilon^n + A = \sigma \tag{2}$$

The effects of thermal softening and strain rate strengthening are neglected. By transforming Equation (2) and dividing Equation (2) by the natural logarithm into both sides, Equation (2) was changed to the following:

$$n\ln\varepsilon + \ln B = \ln(\sigma - A) \tag{3}$$

Figure 6 depicts the relationship of $\ln\varepsilon$ and $\ln(\sigma - A)$ after carrying out the linear fitting by substituting the values of stress and strain into Equation (3). The values of $n$ and $\ln B$, represent the slope and the initial value of the fitting curve, respectively. As a result, the coefficient can be calculated as $n = 1.33$ and $B = 1422.26$ MPa.

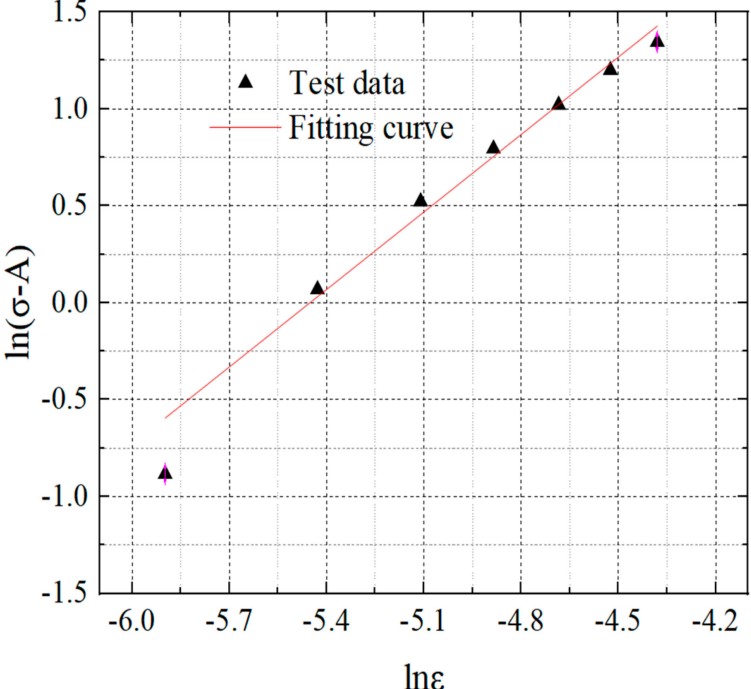

**Figure 6.** The relationship of $\ln\varepsilon$ and $\ln(\sigma - A)$.

### 5.1.2. Identification of Parameter C

Under the deformation temperature in this experiment, $T = T_R = 293$ K, Equation (1) was rearranged as:

$$C\ln\left(\frac{\dot{\varepsilon}}{\dot{\varepsilon}_R}\right) + 1 = \frac{\sigma}{(A + B\varepsilon^n)} \tag{4}$$

Figure 7 depicts the relationship between $\ln(\dot{\varepsilon}/\dot{\varepsilon}_R)$ and $\sigma/(A + B\varepsilon^n)$ after carrying out the linear fitting by substituting 11 strain values and 3 strain rates obtained in this experiment into Equation (4). The value of $C$ represents the slope of the fitting curve. According to the test data of FRP composites, the value of $C$ can be calculated as 0.45.

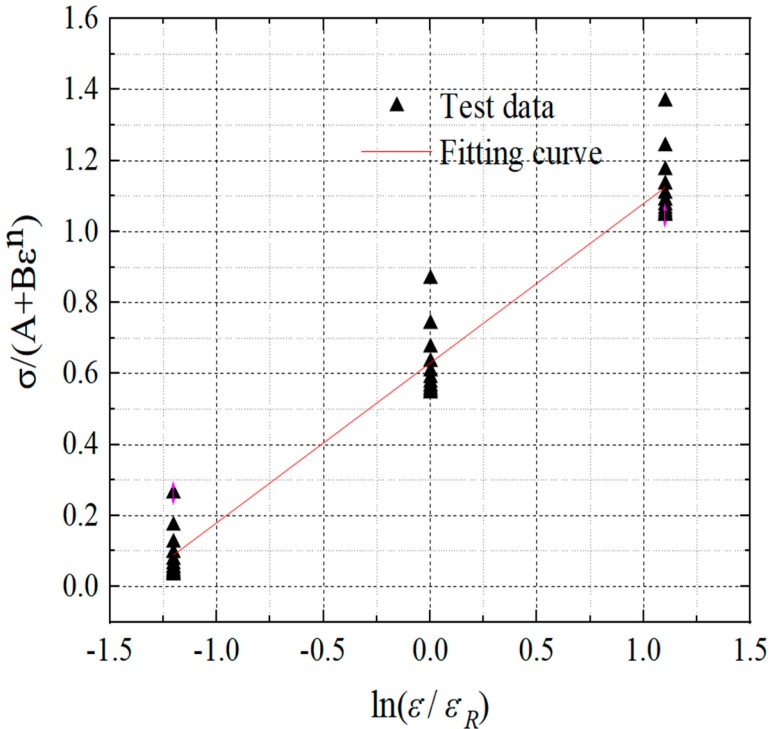

**Figure 7.** The relationship between $\ln(\dot{\varepsilon}/\dot{\varepsilon}_R)$ and $\sigma/(A + B\varepsilon^n)$.

### 5.1.3. Identification of Parameter m

Under the deformation rate in this experiment, $\dot{\varepsilon} = \dot{\varepsilon}_R = 0.005\ \mathrm{s}^{-1}$, Equation (1) was rearranged as:

$$m \ln \frac{T - T_R}{T_m - T_R} = \ln\left[1 - \frac{\sigma}{(A + B\varepsilon^n)}\right] \tag{5}$$

Figure 8 depicts the $\ln[(T - T_R)/(T_m - T_R)] - \ln[1 - \sigma/(A + B\varepsilon^n)]$ curve after carrying out the linear fitting by substituting the 11 strain values and 4 deformation temperatures determined in this study into Equation (5). The value of $m$ represents the slope of the fitting curves. Based on the experimental data of FRP composites, the value of $m$ is 0.45.

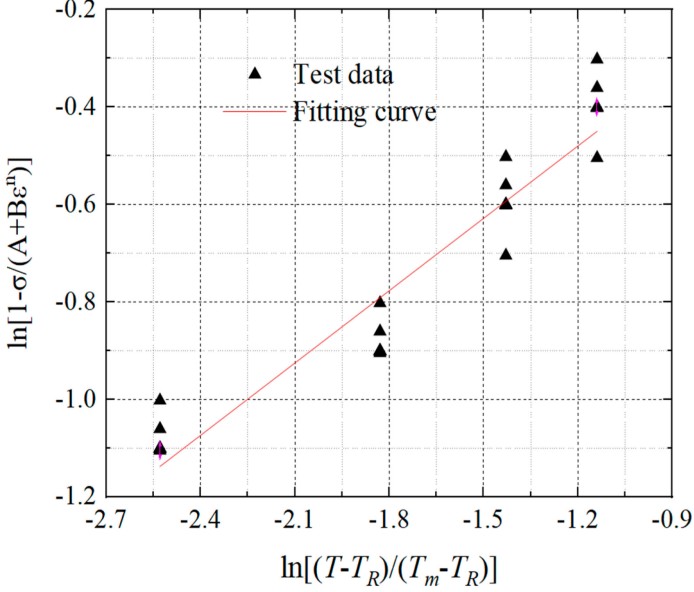

**Figure 8.** The relationship of $\ln[(T - T_R)/(T_m - T_R)]$ and $\ln[1 - \sigma/(A + B\varepsilon^n)]$.

Finally, the relationship among stress $\sigma$, strain $\varepsilon$, deformation temperature $T$ and deformation rate $\dot{\varepsilon}$ was established according to the Johnson–Cook model:

$$\sigma = \left[9.43 + 1422.26 \times \varepsilon^{1.33}\right] \times \left[1 + 0.45 \times \ln \frac{\dot{\varepsilon}}{0.005}\right] \times \left[1 - \left(\frac{T - 293}{1007}\right)^{0.78}\right] - T_0 \quad (6)$$

### 5.2. Verification of the Constitutive Model

The cross-section of the FRP specimens was taken into account as a quantitative parameter throughout the stretching process for the engineering stress−strain curves. This primarily refers to how the cross-section of specimens changed in response to the tensile load. In reality, before tension fracture, the specimen's cross-section steadily declines. The true stress−strain curves of various materials clearly illustrate the impact of the elevated temperatures, where the $\sigma_T$ and $\varepsilon_T$ can be converted by the following expression:

$$\sigma_T = \sigma_E(1 + \varepsilon_E) \quad (7)$$

$$\varepsilon_T = \ln(1 + \varepsilon_E) \quad (8)$$

where $\sigma_E$ represents the engineering stress and $\sigma_T$ represents the true stress, and $\varepsilon_E$ and $\varepsilon_T$ represent the engineering strain and the true strain, respectively.

In this paper, model validation was conducted by comparing true stress−strain curves from experiments with those obtained from computer simulations. The validation was conducted using experimental true stress−strain curves for FRP composites at the deformation rate of 0.005 s$^{-1}$, which were used to establish model parameters. The Johnson−Cook constitutive model for FRP composites at elevated temperatures was used to finally identify the material properties listed in Table 1. Figure 9 represents the comparison between test data and simulated data by the Johnson−Cook constitutive model at elevated temperatures. As seen in Figure 9, some deviation was seen, and the linear assumption was mostly to account for the inaccuracy. The findings were generally satisfactory, indicating that the linear assumption is appropriate and that this proposed Johnson−Cook constitutive model can accurately depict the true stress−strain behavior of FRP composites in the fire scenario.

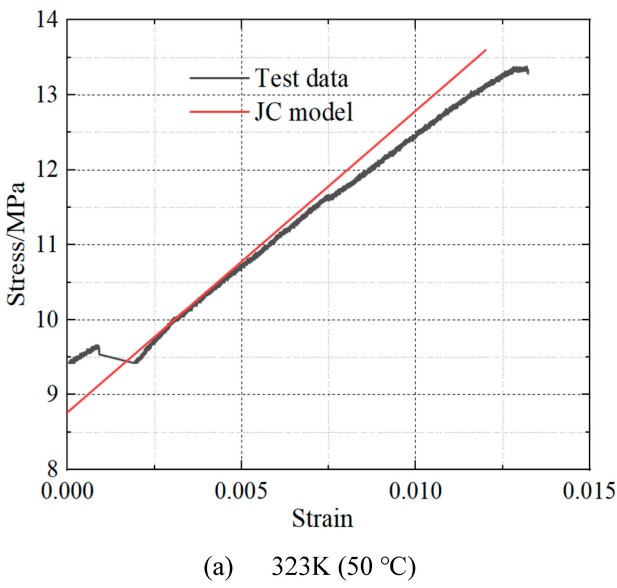

(a)    323K (50 °C)

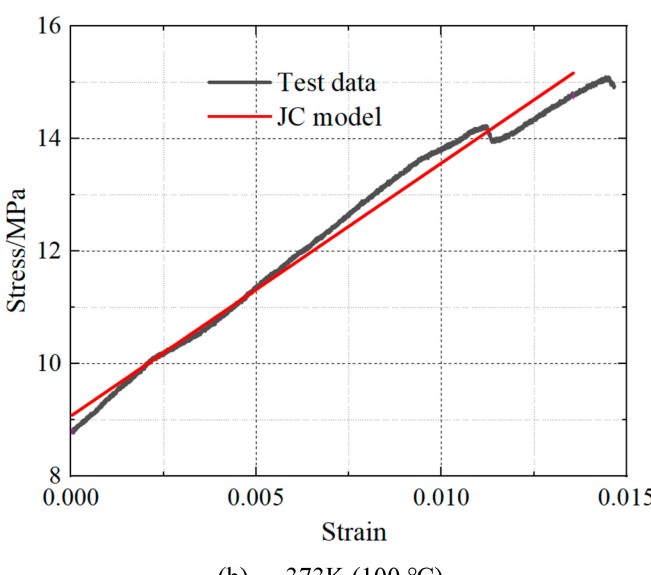

(b)    373K (100 °C)

**Figure 9.** *Cont.*

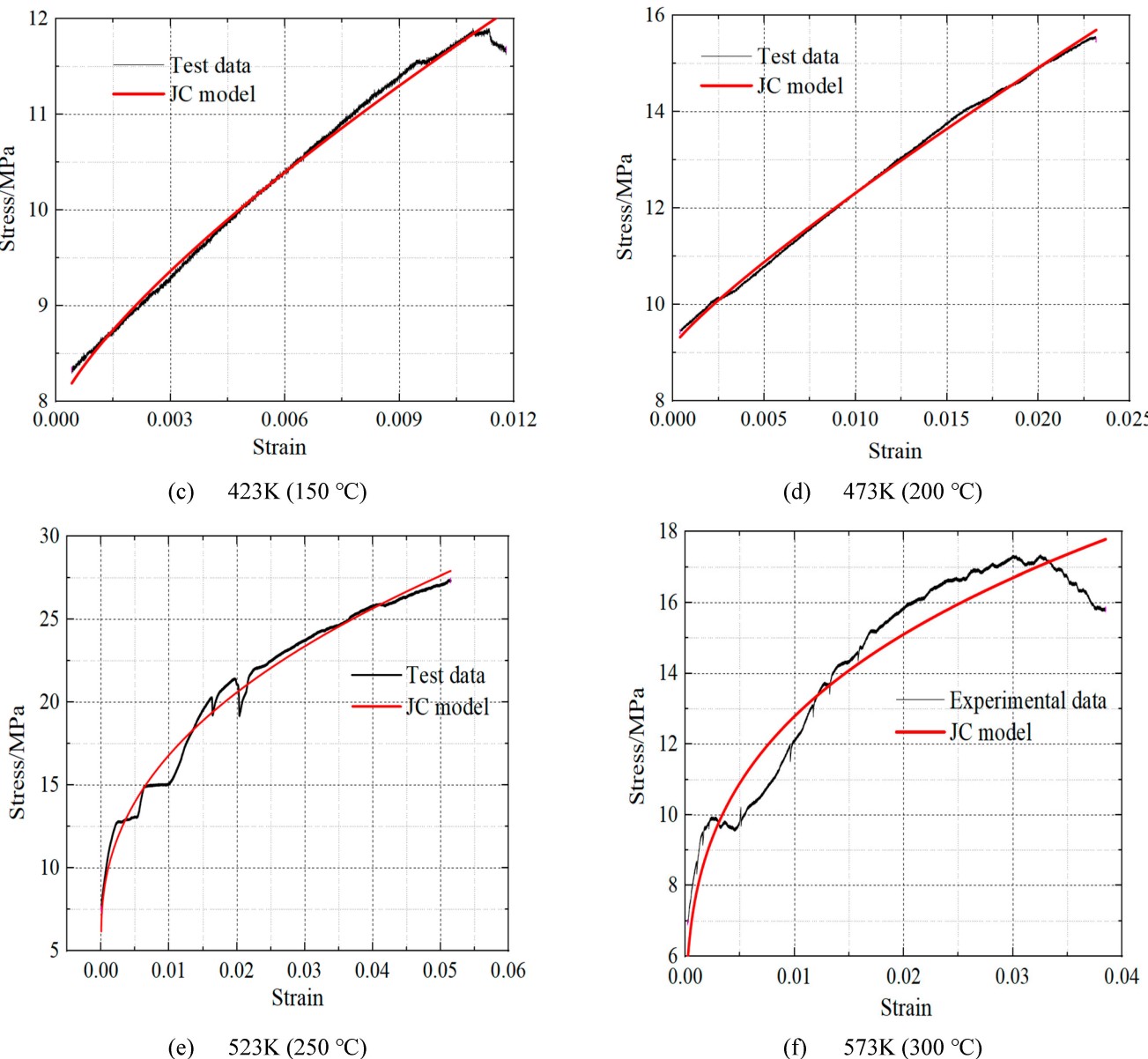

**Figure 9.** Comparison between test data and simulated data by the Johnson−Cook constitutive model at various high temperatures.

## 6. Conclusions

To better comprehend how elevated temperatures affect composites made of fiber, an investigation on the mechanical properties of FRP composites exposed to temperatures of 20–350 °C was experimentally researched. Simultaneously, the FRP specimens were axially loaded until fracture to observe the failure visual observations and mechanical properties. Finally, a new Johnson−Cook constitutive model was proposed to predict the behavior of FRP specimens in the fire scenario. The following are the significant conclusions of this experiment:

1.  The mechanical properties of FRP composites had a critical temperature of 200 °C. When exposed to temperatures below 200 °C, elevated temperatures had a minor influence on the mechanical properties of FRP composites. When exposed to temperatures above 200 °C, the mechanical properties of FRP composites exhibited significant differences.
2.  The ultimate bearing temperature of FRP composites was 300 °C. When exposed to temperatures above 300 °C, the mechanical properties which include ultimate load, fracture load, fracture displacement, and elastic modulus decreased sharply.

3.  The elevated temperatures exerted a significant influence on the surface color of the FRP composites. The surface color of FRP composites gradually changed from fully brown to black with increasing temperatures.
4.  This proposed Johnson−Cook constitutive model can accurately depict the true stress−strain behavior of FRP composites at elevated temperatures.

**Author Contributions:** Conceptualization, Methodology, Investigation, Data curation, Writing—reviewing & editing, C.Z.; Investigation, Data curation, Writing—Original draft preparation, Y.L.; Investigation, J.W. All authors have read and agreed to the published version of the manuscript.

**Funding:** This research was funded by the National Natural Science Foundation of China (grant No. 51508482) and the Natural Science Foundation of Tibet Autonomous (grant No. CGZH2018000014).

**Institutional Review Board Statement:** Not applicable.

**Informed Consent Statement:** Not applicable.

**Data Availability Statement:** The data presented in this study are available on request from the corresponding author. The data are not publicly available due to [Fund Requirements].

**Conflicts of Interest:** The authors declare no conflict of interest.

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
