# Peer review of "Mechanical Properties of Fiber-Reinforced Polymer (FRP) Composites at Elevated Temperatures"

_buildings, doi:10.3390/buildings13010067_

Round 1

Reviewer 1 Report

Paper title:

 Mechanical properties of fiber-reinforced polymer (FRP) composites at elevated temperatures

Authors: Chuntao Zhang, Yanyan Li, Junjie Wu

The work is interesting for the researchers and very useful for the practicing engineers.

In general, the manuscript is well written.

About the abstract. It is well written, however more quantitative comments and concluding remarks about the conclusions drawn from the experimental project would increase the interest of the reader.

Test details are presented in paragraph 2.2. Nevertheless, clarification of the condition during testing is required. The paragraph has to clarify if the temperature of specimens reaches the target temperature and then starts the loading or the loading or the loading increases as the temperature increases.

Considering that specimens after 30 minutes at target temperature they are gradually cooled until environmental temperature (let’s say 20oC) and then the tensile tests take place. A comment about the expected behaviour of the material would be very interesting indeed.

Figures 3a and 3b. Something is wrong between legends and figures.

From figures 3e and 3f it seems that the tensional strength and the ductility capacity of the FRPs significantly increased for target temperatures 250oC and 300oC compared to the ones observed in the ambient temperature. This observation is commented in discussion of paragraph 4.1. Nevertheless, these observations are very important and more clarifying and justifying comments are expected. Perhaps the authors have to re-examine the specifications of the material as described by the manufacturer. In any case, more comments are expected.

A question raises about the FRP strengthening materials after fire. A comment would help.

Stress-strain curves of the materials (as given by the manufacturer or as observed) are not presented in the manuscript. Add stress-strain curves of the materials.

The literature background reported in Introduction is rather informative.

Comments about the observed cracking of the material.

Conclusions are fine.

 Final Conclusion

The submitted paper can be considered as a useful work that could be accepted after a minor revision. 

Author Response

We appreciate very much the time and effort that the reviewer spent on this paper. We have carefully read and considered the comments made by the reviewer, and these comments are valuable to improve the quality manuscript. According to these comments, we conducted a detailed revision. All the comments are addressed, and the revised contents are printed in red in the text. All changes concerning the comments or answers to the questions are shown below.

Reviewer: #1: The work is interesting for the researchers and very useful for the practicing engineers.In general, the manuscript is well written.

Thank you a lot for your invaluable comments.

  1. About the abstract. It is well written, however more quantitative comments and concluding remarks about the conclusions drawn from the experimental project would increase the interest of the reader.

Reply: Thanks for your helpful suggestion. Undoubtedly, more quantitative comments and concluding remarks about the conclusions drawn from the experimental project would increase the interest of the reader. Some comments and remarks are added in the paper to clarify this issue as: 

The results demonstrate that the exposure temperatures have a major impact on the residual mechanical properties of fiber-reinforced polymer (FRP) composites when the exposed temperatures exceed 200℃. Below 200℃, the maximum decrease and increase in the fracture load of fiber reinforced polymer (FRP) composites were between −34% and 153% of their initial fracture load. After exposing to temperatures above 200℃, the surface color of fiber reinforced polymer (FRP) composites changed from brown to black. When exposed to temperatures between 200-300℃, the ultimate load of fiber reinforced polymer (FRP) composites significantly increased from 731.01N to 1650.97N.

  1. Test details are presented in paragraph 2.2. Nevertheless, clarification of the condition during testing is required. The paragraph has to clarify if the temperature of specimens reaches the target temperature and then starts the loading or the loading or the loading increases as the temperature increases.

Reply: Thanks for your comments. We mentioned that the temperature of specimens reaches the target temperature: To ensure uniform temperature over the entire gauge length, the specimen was kept for about 20 min at this temperature. However, we did not describe “then starts the loading and the loading increases as the temperature increases”. Some comments and remarks are added in the paper to clarify this issue as:

  Then the specimen was loaded until it fails, during which the tensile strain rate is constant and the loading increases as the temperature increases.

  1. Considering that specimens after 30 minutes at target temperature they are gradually cooled until environmental temperature (let’s say 20℃) and then the tensile tests take place. A comment about the expected behaviour of the material would be very interesting indeed.

Reply: Thank you a lot for your comments. Specimens after 30 minutes at target temperature, they were still subjected to elevated temperatures and would not cool until environmental temperature. Some comments and remarks are added in the paper to clarify this issue as:

To ensure uniform temperature over the entire gauge length, the specimen was kept for about 30 min at this temperature and the elevated temperatures still remained unchanged.

  1. Figures 3a and 3b. Something is wrong between legends and figures.

Reply: Thanks for your comments. Figures 3a and 3b shows the incorrect legends and figures of comparison between ambient temperature and target temperature due to my negligence. We have corrected the incorrect legends and figures in this paper.

  1. From figures 3e and 3f it seems that the tensional strength and the ductility capacity of the FRPs significantly increased for target temperatures 250℃ and 300℃ compared to the ones observed in the ambient temperature. This observation is commented in discussion of paragraph 4.1. Nevertheless, these observations are very important and more clarifying and justifying comments are expected. Perhaps the authors have to re-examine the specifications of the material as described by the manufacturer. In any case, more comments are expected.

Reply: Thanks for your helpful comments. Yes, the tensional strength and the ductility capacity of the FRPs are very important, the tensional strength are discussed in paragraph 4.1 and the ductility are discussed in paragraph 4.3 (fracture displacement) since the fracture displacement reflects ductility in some extent. Some modifications are as follows:

However, the tensile load-displacement curves of FRP composites decreased sharply when exposed to temperatures of 350℃, which indicated that the tensional strength and ductility increased significantly at those target temperatures, as shown in Fig.3(e) and Fig.3(f).

  1. Stress-strain curves of the materials (as given by the manufacturer or as observed) are not presented in the manuscript. Add stress-strain curves of the materials.

Reply: Thanks for your suggestion. Due to the small value of load in the test, if the stress-strain curves are used to describe FRPs, the change is not obvious. Therefore, in this paper, we use the load-displacement curves to replace the stress-strain curves.

  1. Comments about the observed cracking of the material.

Reply: Thank you a lot for your suggestion. In this paper ,we only discussed the color change of FRPs after test. However, the observed cracking of the material is also important. Some descriptions are added as follows:

It is worth noting that the fibers on the surface of the fiber composite material were shed after exposure to temperatures above 200℃. Moreover, with the increase of temperature, the phenomenon of spalling at the center fracture position of fiber-reinforced polymer (FRP) composites was more obvious. 

Reviewer 2 Report

1- abstract need to rewrite. Mention the novelty and originalncontribution to the existing knowledge. Mention outcome of this study.

2- add state of the art previous studies on the conducted experimental work. Cite latest papers on that. Identify research gap.

3- put clear pictures of tested specimens, fig 2 need to be replaced with high quality pictures.

4- can number of samples be increased?

5- can u add chemical composition of tested samples 

6- also if possible add micro-structural images of tested samples.

7- can u compare your results with other people findings 

8- put a separate section for discussion.

9- references style need to be in accordance with journal guidelines.

Author Response

Response to Comments from Reviewer No. 2:

We appreciate very much the time and effort that the reviewer spent on this paper. We have carefully read and considered the comments made by the reviewer, and these comments are valuable to improve the quality manuscript. Based on these comments, we conducted a detailed revision. All the comments are addressed, and the revised contents are printed in red in the text. All changes concerning the comments or answers to the questions are shown below.

Reviewer #2: Abstract need to rewrite. Mention the novelty and original contribution to the existing knowledge. Mention outcome of this study.

Reply: Thanks for your comments. The novelty and original contribution to the existing knowledge have descriped in the section 1 due to the brevity of the abstract. We have added the outcome of this study and the descriptions are as follows:

The experimental results studied in this research can be applicable to both further research and engineering applications when conducting a theoretical simulation of fiber reinforced polymer (FRP) composites.  

1.Add state of the art previous studies on the conducted experimental work. Cite latest papers on that. Identify research gap.

Reply: Thank you a lot for your comments. According to previous studies, a significant decrease or increase is exhibited in the post-fire mechanical properties of various steels. However, the mechanical properties of FRPs at elevated temperatures have not been researched yet. Addionally, the constituve model of FRPs at elevated temperatures have also not been established. So the research gap of the FRPs is obvious. Some descriptions are added in the paper to clarify this issue as:

Only a few studies on fiber-reinforced polymer (FRP) composites at high temperatures have been conducted. The bonding strength of the concrete matrix between carbon and glass fiber sheet changes after being exposed to temperatures of 20, 50, 65, and 80 °C, respectively, according to research by Leone M [6]. And the outcomes showed that the concrete matrix's transition temperature from shearing failure to cohesion failure was 65 °C. Salloum [8] conducted an axial compression test on FRP-strengthened cylinders after exposing them to temperatures of 100 and 200 °C, and they were left at each target temperature for 1, 2, and 3 hours, respectively. The test specimens' diameters were Φ100 × 200 mm. The test findings indicated that external-bonded FRP materials' reinforcing efficacy was considerably sensitive to high temperatures. At temperatures 2.5 times Tg, the ultimate capacity of concrete specimens enhanced with FRP was 25% less than it was at ambient temperatures. After being exposed to 120, 130, 150, and 180 °C, respectively, GFRP-strengthened concrete cylinders' ultimate axial compressive strength was reduced by 2%, 4%, 13%, and 18% [9]. However, at 150 and 185 °C, the ultimate compressive strength of GFRP-strengthened concrete cylinders with an epoxy-based fireproofing coating reduced by around 3% and 10%. At various temperatures, the failure mechanisms of GFRP-strengthened concrete cylinders have been identified as fiber-dominated at lower temperatures and resin-dominated at higher temperatures. Chowdhury [10] has studied how full-scale reinforced concrete cylinders (400 3810 mm) with exterior fire insulation that contains an FRP-reinforced layer alter in structural qualities when exposed to fire. The interior reinforced bar and concrete of FRP-wrapped reinforced concrete buildings maintained a lower temperature for up to 300 minutes due to the exterior fire insulation. Nevertheless, during 34 minutes, the temperature of the FRP-reinforced layer insulated by 53 mm of fire insulation stayed below its Tg.

2.Put clear pictures of tested specimens, fig 2 need to be replaced with high quality pictures.  

Reply: Thanks for your suggestions. As you suggested, I have replaced the picture with high quality pictures.

3.Can number of samples be increased?

Reply: Thanks for your suggestions. At each test condition, we have used three specimens to reduce the test error and a total of 27 specimens were designed.

4.Can u add chemical composition of tested samples?

Reply: Thanks for your suggestions. We donnt have the specific chemical compositions, but we have the description of the specimens in this test. Some descriptions are added in the paper to clarify this issue as:

Nano-montmorillonite composite fiberboard with nano-montmorillonite modified biomass resin as matrix, basalt fiber as reinforcement material, plate surface attached with silicon nitride coating, including 1) basalt fiber is natural basalt ore as raw material, the broken after 1500℃ melting, through platinum rhodium alloy wire drawing leakage plate made of a new type of inorganic environmental protection green high performance fiber material; 2) Nanometer montmorillonite resin from montmorillonite raw materials, through intercalation, modification, stripping treatment to obtain montmorillonite with a slice diameter of less than 100nm, the adsorption effect is 3 times that of ordinary montmorillonite products; 3) Silicon nitride coating is a compound of nitrogen and silicon, is a kind of superhard substance, wear resistance, high temperature resistance, corrosion resistance, oxidation resistance, lubrication, self-cleaning and other characteristics, mostly used in the military field.

5.Also if possible add micro-structural images of tested samples.

Reply: Thanks for your suggestions. The test specimens of FRPs after fracture cannot be used in the fracture scanning machine since the machine can only test metal blocks.

6.Can u compare your results with other people findings?

Reply: Thanks for your suggestions. Since we have modified the introduction part and added the other people findings of structure components of FRPs, we donot compare the test results due to our research is dealing with structural components and not composite structures.

7.Put a separate section for discussion..

Reply: Thank you a lot for your invaluable comments. We have modified the separate section for discussion to study the effects of elevated temperatures on fracture load, ultimate load, fracture displacement and elastic modulus.

8.References style need to be in accordance with journal guidelines

Reply: Thanks for your suggestions. We have modified trhe references style which are in accordance with journal guidelines. Some references are as follows:

  • M. Rowell, Natural fibers: types and properties, Wood head publishing Series in Composites Science and Eng. (2008) 3-66.
  • Lange, R. Schneider, Constitutive equations of structural steel S460 at high temperatures, J. Struct. Fire Eng. 2 (3) (2009) 217–230.
  • Zhang, R. Wang, G. Song, Post-fire mechanical properties of Q460 and Q690 high strength steels after fire-fighting foam cooling, Thin-Walled Struct. 156 (2020) 106983.
  • Zhang, R. Wang, L. Zhu, Mechanical properties of Q345 structural steel after artificial cooling from elevated temperatures, J. Construct. Steel Res. 176 (2021) 106432.

Reviewer 3 Report

The study presented an experimental evaluation on the effect of elevated temperature on the mechanical properties of FRP materials. An empirical equation was then proposed to describe the mechanical properties of the investigated composites under elevated temperature. The study presents a relatively good set of test data that can contribute to the understanding of composites in elevated temperature. However, the discussion is mostly limited to trend analysis but with very limited discussion of the mechanism on how the elevated temperature affects the mechanical properties. In addition, most of the section are very cursory and require strong improvements. In its current form, I strongly suggest that the authors revise the manuscript to highlight the significant and new findings from the work. In addition to improving the analysis and discussion, the following comments/suggestions should be addressed:

1)      The introduction section must be completely re-organized. The state of the art about the variation with temperature of the FRP mechanical properties is almost null (although several references are available in the literature about this topic). Most of the information provided in section 1 are completely out of context (natural fibres, steel etc) and can distract the readers from the main topic of your paper: MECHANICAL PROPERTIES OF FRP and TEMEPERATURE! Below you can find some general references about your topic:

DOI: https://doi.org/10.1016/j.conbuildmat.2022.128340

DOI: 10.2749/guimaraes.2019.0861

DOI: 10.1007/s10694-009-0116-6

DOI: https://doi.org/10.1016/j.conbuildmat.2019.01.003

But please, make an additional effort to include also other references which are available in the literature concerning the influence of the temperature on FRP materials.

2)      At the end of the introduction section, the aim of study should be more emphasized.

3)      Section 2.1. must be significantly improved, at this stage it does not provide useful information for the reader. No information about the type of fibres and resin used in the study are given. No information about the thermo-mechanical properties (glass transition temperature and decomposition temperature) of the FRP constituent materials are provided. Finally, no information regarding the composite layup sequence is reported.  All this information is crucial to better understand the variation with temperature of the FRP mechanical properties.

4)      Section 2.1. should report only the information about the materials and test specimens. General info should be only provided in the Introduction section. For instance, lines 97 to 104 should be deleted.

5)      Section 2.2 should be also improved. The heating procedure and loading procedure must be described in more details. In addition, from the text, it is not clear if the target temperature was defined based on the temperature reached by the specimens or by the air inside the furnace. Please specify.

6)      Section 2.2, please specify how strains were measured during the tests.

7)      Section 3, the results obtained at 250 and 300 C° are difficult to understand. The authors should provide some “strong” explanation to justify them. In addition, the discussion is mostly limited to trend analysis but with very limited discussion of the mechanism on how the elevated temperature affects the mechanical properties.

8)     In the conclusions, the authors should explain the significance and shortcomings of the research work, instead of repeating the results obtained before.

Author Response

We appreciate very much the time and effort that the reviewer spent on this paper. We have carefully read and considered the comments made by the reviewer, and these comments are valuable to improve the quality manuscript. Based on these comments, we conducted a detailed revision. All the comments are addressed, and the revised contents are printed in red in the text. All changes concerning the comments or answers to the questions are shown below.

Reviewer #3: The study presented an experimental evaluation on the effect of elevated temperature on the mechanical properties of FRP materials. An empirical equation was then proposed to describe the mechanical properties of the investigated composites under elevated temperature. The study presents a relatively good set of test data that can contribute to the understanding of composites in elevated temperature. However, the discussion is mostly limited to trend analysis but with very limited discussion of the mechanism on how the elevated temperature affects the mechanical properties. In addition, most of the section are very cursory and require strong improvements. In its current form, I strongly suggest that the authors revise the manuscript to highlight the significant and new findings from the work.

Reply: Thank you a lot for your invaluable comments. Because of our writing problems, you question the limited discussion of the mechanism on how the elevated temperature affects the mechanical properties and suggest us to revise the manuscript to highlight the significant and new findings from the work. This research aims to study the residual mechanical properties of FRPs at elevated temperatures and further estanlish the constitutive model to predict the stress-strain bebavior of FPRs at fire. We have added the descriptions about the mechanism on how the elevated temperature affects the mechanical properties.Some descriptions are added in the paper to clarify this issue as:

The mechanical properties of fiber-reinforced resin matrix composites are sensitive to temperature, and the fire resistance of reinforced fiber is better than that of the resin matrix. Therefore, the mechanical properties of FRP at high temperatures are mainly controlled by the resin matrix. The matrix resin of FRP includes thermosetting resin and thermoplastic resin. In the fire environment, with the increase in temperature, the resin matrix is softened and enters the rubber state from the glass state. The ability of the resin matrix to transfer shear stress between reinforced fibers is reduced. In addition to the temperature characteristic of the matrix resin, the fire reaction characteristic of the matrix resin itself is also important to the high-temperature mechanical properties of the composites. Under the action of high temperature, the resin matrix can still form a composite effect with the reinforced fiber to some extent.

In addition, there are several problems as follows:

  1. The introduction section must be completely re-organized. The state of the art about the variation with temperature of the FRP mechanical properties is almost null (although several references are available in the literature about this topic). Most of the information provided in section 1 are completely out of context (natural fibres, steel etc) and can distract the readers from the main topic of your paper: MECHANICAL PROPERTIES OF FRP and TEMEPERATURE! Below you can find some general references about your topic:

DOI: https://doi.org/10.1016/j.conbuildmat.2022.128340

DOI: 10.2749/guimaraes.2019.0861

DOI: 10.1007/s10694-009-0116-6

DOI: https://doi.org/10.1016/j.conbuildmat.2019.01.003

But please, make an additional effort to include also other references which are available in the literature concerning the influence of the temperature on FRP materials..

Reply: Thank you a lot for your invaluable comments. The descriptions about the natural fibers aim to introduce the fiber-reinforced polymer (FRP) composites in our test, and the descriptions about the steels were designed to clarify the importance of considering natural hazards (fire) since the research about the FRPs at elevated temperatures are limited. We have modified the introduction section and deleted some descriptions about the steels and added some references about the mechanical properties of FRPs components or structures at elevated temperatures based on your suggestion. Some descriptions are added in the paper to clarify this issue as:

Only a few studies on fiber-reinforced polymer (FRP) composites at high temperatures have been conducted. The bonding strength of the concrete matrix between carbon and glass fiber sheet changes after being exposed to temperatures of 20, 50, 65, and 80 °C, respectively, according to research by Leone M [6]. And the outcomes showed that the concrete matrix's transition temperature from shearing failure to cohesion failure was 65 °C. Salloum [8] conducted an axial compression test on FRP-strengthened cylinders after exposing them to temperatures of 100 and 200 °C, and they were left at each target temperature for 1, 2, and 3 hours, respectively. The test specimens' diameters were Φ100 × 200 mm. The test findings indicated that external-bonded FRP materials' reinforcing efficacy was considerably sensitive to high temperatures. At temperatures 2.5 times Tg, the ultimate capacity of concrete specimens enhanced with FRP was 25% less than it was at ambient temperatures. After being exposed to 120, 130, 150, and 180 °C, respectively, GFRP-strengthened concrete cylinders' ultimate axial compressive strength was reduced by 2%, 4%, 13%, and 18% [9]. However, at 150 and 185 °C, the ultimate compressive strength of GFRP-strengthened concrete cylinders with an epoxy-based fireproofing coating reduced by around 3% and 10%. At various temperatures, the failure mechanisms of GFRP-strengthened concrete cylinders have been identified as fiber-dominated at lower temperatures and resin-dominated at higher temperatures. Chowdhury [10] has studied how full-scale reinforced concrete cylinders (400 3810 mm) with exterior fire insulation that contains an FRP-reinforced layer alter in structural qualities when exposed to fire. The interior reinforced bar and concrete of FRP-wrapped reinforced concrete buildings maintained a lower temperature for up to 300 minutes due to the exterior fire insulation. Nevertheless, during 34 minutes, the temperature of the FRP-reinforced layer insulated by 53 mm of fire insulation stayed below its Tg.

Generally speaking, the matrix resin of FRP includes thermosetting resin and thermoplastic resin. In the fire environment, with the increase of temperature, the mechanical properties of composite materials mainly experience a three times decrease. When the temperature rises to the glass transition temperature Tg of the resin matrix, the resin matrix softens and enters the rubber state from the glass state. The ability of the resin matrix to transfer shear stress between reinforced fibers decreases, resulting in the first significant decrease in the mechanical properties of FRP. When the temperature is further raised to the resin decomposition temperature Td (about 300 ~ 400℃), the matrix of FRP is gradually decomposed and carbonized, and the toxic smoke is released, resulting in the second significant decrease in the mechanical properties of FRP. At higher temperature, the resin matrix begins to burn, the combustion process will release more heat, and release toxic smoke, resulting in the second significant decrease in the mechanical properties of FRP; At higher temperatures, the resin matrix begins to burn, releasing more heat during the combustion process.

  1. At the end of the introduction section, the aim of study should be more emphasized.

Reply: Thank you a lot for your invaluable suggestions. We have added some descriptions about the aim of the study at the end of the introduction section. Some descriptions are as follows:

To fill the gap, this research conducted an experimental study of FRP composites at high temperatures ranging from 20 °C to 350 °C and proposed a constitutive JC model which considers the impact of elevated temperatures. The experimental results studied in this research can be applicable to both further research and engineering applications when conducting a theoretical simulation of fiber-reinforced polymer (FRP) composites.

  1. Section 2.1. must be significantly improved, at this stage it does not provide useful information for the reader. No information about the type of fibres and resin used in the study are given. No information about the thermo-mechanical properties (glass transition temperature and decomposition temperature) of the FRP constituent materials are provided. Finally, no information regarding the composite layup sequence is reported. All this information is crucial to better understand the variation with temperature of the FRP mechanical properties.

Reply: Thank you a lot for your invaluable comments. We have added information about the type of fibres and resin used in the study and the thermo-mechanical properties (glass transition temperature and decomposition temperature) of the FRP constituent materials. However, the information was put into the introduction part. Some descriptions are added in the paper to clarify this issue as:

In this test, Nano-montmorillonite composite fiberboard with nano-montmorillonite modified biomass resin as matrix, basalt fiber as reinforcement material, plate surface attached with silicon nitride coating, including 1) basalt fiber is natural basalt ore as raw material, the broken after 1500℃ melting, through platinum-rhodium alloy wire drawing leakage plate made of a new type of inorganic environmental protection green high performance fiber material; 2) Nanometer montmorillonite resin from montmorillonite raw materials, through intercalation, modification, stripping treatment to obtain montmorillonite with a slice diameter of less than 100nm, the adsorption effect is 3 times that of ordinary montmorillonite products; 3) Silicon nitride coating is a compound of nitrogen and silicon, is a kind of super-hard substance, wear resistance, high temperature resistance, corrosion resistance, oxidation resistance, lubrication, self-cleaning and other characteristics, mostly used in the military field.

  1. Section 2.1. should report only the information about the materials and test specimens. General info should be only provided in the Introduction section. For instance, lines 97 to 104 should be deleted.

Thanks a lot for your helpful comments. We have already deleted lines 97 to 104 and put the general information in the introduction part.

  1. Section 2.2 should be also improved. The heating procedure and loading procedure must be described in more details. In addition, from the text, it is not clear if the target temperature was defined based on the temperature reached by the specimens or by the air inside the furnace. Please specify.

Reply: Thank you a lot for your invaluable comments. The target temperature is controlled by the machine, we just need to set the data. The heating procedure and loading procedure are described in more detail. Some descriptions are as follows:

To ensure uniform temperature over the entire gauge length, the specimen was kept for about 30 min at this temperature and the elevated temperatures still remained unchanged. Then the specimen was loaded until it fails, during which the tensile strain rate is constant and the loading increases as the temperature increases.

  1. Section 2.2, please specify how strains were measured during the tests.

Reply: Thanks a lot for your helpful comments. In this test, we adopted displacement control and it is controlled by the test machine, we just need to set the data. Some descrioptions are as follows:

The displacement control method was used to test the specimens at a constant rate of 1 mm/min until fracture, which conformed to the requirements of GB/T 228.1-2010.

  1. Section 3, the results obtained at 250 and 300 C° are difficult to understand. The authors should provide some “strong” explanation to justify them. In addition, the discussion is mostly limited to trend analysis but with very limited discussion of the mechanism on how the elevated temperature affects the mechanical properties.

Reply: Thank you a lot for your invaluable comments. Due to the changes of diverse mechanical properties at temperatures of 250℃ and 300℃, the results are obtained. Because of our writing problems, you question the limited discussion of the mechanism of how the elevated temperature affects the mechanical properties We have added the descriptions about the mechanism on how the elevated temperature affects the mechanical properties.Some descriptions are added in the paper to clarify this issue as:

The reason for this phenomenon is that the matrix bonded by the fiber resin changes with the increase of temperature, which the glue or epoxy resin is softened at high temperatures. Furthermore, the mechanical properties of FRP at high temperatures are mainly controlled by the resin matrix. The matrix resin of FRP includes thermosetting resin and thermoplastic resin. In the real fire situation, with the increase of temperature, the resin matrix is softened and enters the rubber state from the glass state. The ability of the resin matrix to transfer shear stress between reinforced fibers is reduced. In addition to the temperature characteristic of the matrix resin, the fire reaction characteristic of the matrix resin itself is also significant to the high temperatures mechanical properties of the composites. Under the effect of high temperature, the resin matrix can still form a composite effect with the reinforced fiber to some extent.

  1. In the conclusions, the authors should explain the significance and shortcomings of the research work, instead of repeating the results obtained before.

Reply: Thanks a lot for your helpful comments. Since the conclusion of most articles is a summary of the previous findings, we also adopt this way of writing. The significance and shortcomings of the research work are as follows:

Significance: The experimental results studied in this research can be applied to both further research and engineering applications when conducting a theoretical simulation of fiber-reinforced polymer (FRP) composites.

Shortcomings: In order to simulate the real fire scenario, more elevated temperatures are required.

Round 2

Reviewer 1 Report

Revised Paper title:

 Mechanical properties of fiber-reinforced polymer (FRP) composites at elevated temperatures

Authors: Chuntao Zhang, Yanyan Li, Junjie Wu

As stated in the first review the work is interesting for the researchers and very useful for the practicing engineers and in general, the manuscript is well written.

About the abstract. In the revised manuscript quantitative comments and concluding remarks have been added in the abstract, as recommended by the reviewer.

Clarifying comments about the conditions during testing have been added, too.

 More or less, the authors have successfully responded to the reviewer’s comments

The literature background reported in Introduction is rather informative.

Conclusions are fine.

 Final Conclusion

The submitted paper is a useful work.  Acceptance is recommended. 

Author Response

Response to Comments from Reviewer No. 1:

We appreciate very much the time and effort that the reviewer spent on this paper. We have carefully read and considered the comments made by the reviewer, and these comments are valuable to improve the quality manuscript. According to these comments, we conducted a detailed revision. All the comments are addressed, and the revised contents are printed in red in the text. All changes concerning the comments or answers to the questions are shown below.

Reviewer: #1: As stated in the first review the work is interesting for the researchers and very useful for the practicing engineers and in general, the manuscript is well written. About the abstract. In the revised manuscript quantitative comments and concluding remarks have been added in the abstract, as recommended by the reviewer.

Clarifying comments about the conditions during testing have been added, too.

 More or less, the authors have successfully responded to the reviewer’s comments

The literature background reported in Introduction is rather informative.

Conclusions are fine.

 Final Conclusion

The submitted paper is a useful work.  Acceptance is recommended. 

Thank you a lot for your invaluable comments. According to your suggestions, we have modified all the mentioned comments again. In the reviewed manuscript, we have reorganized the introduction of this reviewed manuscript. The modifications are as follows:

The above scholars mainly focused on the mechanical behavior of concrete elements reinforced with FRP. There are studies about the mechanical behavior of FRP as a standalone material at elevated temperatures. Pultruded carbon fiber reinforced polymer (P-CFRP) specimens and CFRP tensile specimens manufactured with a hand lay-up method were subjected to a series of tests by Nguyen et al. [3,4] at temperatures that reached 700 °C. According to their findings, hand layup specimens' ultimate tensile strength and Young's modulus were reduced by 50% at 350 °C and 30% at 600 °C, respectively. Additionally, they demonstrated that the thermomechanical strength is lower than the residual strength for P-CFRP samples at the same degree of applied temperature. One of the pioneering studies regarding the behavior and characteristics of FRP materials at high temperatures that are utilized in industrial domains, such as the automotive, marine, and aerospace industries, was performed by Mouritz and Mathys [5]. At high temperatures, Shenghu and Zhishen [6] performed a series of tension tests on single-layer FRP sheets composed of GFRP, CFRP, and basalt-fiber reinforced (BFRP). Among all the tested fiber-reinforced sheets, they concluded that the CFRP sheets had the highest strength and the GFRP sheets had the lowest strength [7]. At around 55°C, all of the sheets' tensile strength significantly decreased, but no further substantial decline occurred as the temperature increased. The CFRP sheets had the highest residual strength, with almost 69% of their initial tensile strength. However, there still lacks the work of establishing the constitutive model to better predict the mechanical behavior of FRPs at elevated temperatures. In this research, we proposed a constitutive model based on the experimental results of FRPs at elevated temperatures to fill the research gap.

We have added the research information about the mechanical properties of FRP at elevated temperatures, especially the mechanical behavior of FRP as a standalone material at high temperatures. The modifications are as follows:

The above scholars mainly focused on the mechanical behavior of concrete elements reinforced with FRP. There are studies about the mechanical behavior of FRP as a standalone material at elevated temperatures. Pultruded carbon fiber reinforced polymer (P-CFRP) specimens and CFRP tensile specimens manufactured with a hand lay-up method were subjected to a series of tests by Nguyen et al. [3,4] at temperatures that reached 700 °C. According to their findings, hand lay-up specimens' ultimate tensile strength and Young's modulus were reduced by 50% at 350 °C and 30% at 600 °C, respectively. Additionally, they demonstrated that the thermomechanical strength is lower than the residual strength for P-CFRP samples at the same degree of applied temperature. One of the pioneering studies regarding the behavior and characteristics of FRP materials at high temperatures that are utilized in industrial domains, such as the automotive, marine, and aerospace industries, was performed by Mouritz and Mathys [5]. At high temperatures, Shenghu and Zhishen [6] performed a series of tension tests on single-layer FRP sheets composed of GFRP, CFRP, and basalt-fiber reinforced (BFRP). Among all the tested fiber-reinforced sheets, they concluded that the CFRP sheets had the highest strength and the GFRP sheets had the lowest strength [7]. At around 55°C, all of the sheets' tensile strength significantly decreased, but no further substantial decline occurred as the temperature increased. The CFRP sheets had the highest residual strength, with almost 69% of their initial tensile strength.

In this research, we conducted an experimental study of FRP composite as a standalone material at high temperatures ranging from 20 °C to 350 °C and proposed a constitutive JC model which considers the impact of elevated temperatures.

Some studies have provided experimental results on the mechanical behaviour of FRP at elevated temperatures. However, most of the tested specimens are FRP bars and a few are FRP plates. Meanwhile, the existing research results have found that the composition of the FRP material, the type of bonding colloid, and the manufacturing process have different effects on the high-temperature mechanical behaviour of FRP materials. Therefore, we have added the research works about the influence of elevated temperature on FRP materials and have highlighted how the work improves the current state of the art. The modifications are as follows:

There are studies about the mechanical behavior of FRP as a standalone material at elevated temperatures. Pultruded carbon fiber reinforced polymer (P-CFRP) specimens and CFRP tensile specimens manufactured with a hand lay-up method were subjected to a series of tests by Nguyen et al. [3,4] at temperatures that reached 700 °C. According to their findings, hand lay-up specimens' ultimate tensile strength and Young's modulus were reduced by 50% at 350 °C and 30% at 600 °C, respectively. Additionally, they demonstrated that the thermomechanical strength is lower than the residual strength for P-CFRP samples at the same degree of applied temperature. One of the pioneering studies regarding the behavior and characteristics of FRP materials at high temperatures that are utilized in industrial domains, such as the automotive, marine, and aerospace industries, was performed by Mouritz and Mathys [5]. At high temperatures, Shenghu and Zhishen [6] performed a series of tension tests on single-layer FRP sheets composed of GFRP, CFRP, and basalt-fiber reinforced (BFRP). Among all the tested fiber-reinforced sheets, they concluded that the CFRP sheets had the highest strength and the GFRP sheets had the lowest strength [7]. At around 55°C, all of the sheets' tensile strength significantly decreased, but no further substantial decline occurred as the temperature increased. The CFRP sheets had the highest residual strength, with almost 69% of their initial tensile strength. However, there still lacks the work of establishing the constitutive model to better predict the mechanical behavior of FRPs at elevated temperatures. In this research, we proposed a constitutive model based on the experimental results of FRPs at elevated temperatures to fill the research gap.

The test specimen in this research was provided by China Southwest Architecture Design and Research Institute and we have obtained the material characteristics of FRPs recently. The thermos-mechanical properties of the FRP material including ultimate load, fracture load, fracture displacement and elastic modulus are discussed in the reviewed manuscript. The following are its detailed material characteristics at ambient temperature:

In this test, physical and mechanical properties of nanometer montmorillonite com-posite fiber material, including the density ρ, Barcol hardness, fiber volume fraction, in-soluble content of resin, water absorption, glass transition temperature Tg, tensile strength (main fiber direction) ƒtm, tensile strength (secondary fiber direction) ƒts, compressive strength (main fiber direction) ƒcm, compressive strength (secondary fiber direction) ƒcs and shock resistance of nanometer montmorillonite composite fiber material are provided in Table 1.

Table 1 Physical and mechanical properties of nanometer montmorillonite composite fiber material.

Performance

Performance index

ρ / (kg•m-3)

≤ 2000

Barcol hardness / (HBa)

≥ 50

Fiber volume fraction / %

≥ 70

Insoluble content of resin / %

≥ 90

Water absorption / %

≤ 1.0

Tg / ℃

≥ 290

ƒtm / MPa

≥ 400

ƒts / MPa

≥ 10

ƒcm / MPa

≥ 100

ƒcs / MPa

≥ 15

Shock resistance / (kJ•m-2)

≥ 240

In the original manuscript, we reported less information about the reason for explaining the non-monotonic variation trend. In the reviewed manuscript, we have modified the descriptions about the mechanism of how the elevated temperature affects the mechanical properties based on the test results. Besides, all the test procedures are carried out following the specifications, and the test results are accurate. But the exact reason that can accurately account for the change is unclear, we are discussing the results of the test. Some descriptions are added in the paper to clarify this issue:

Ultimate load: However, when exposed to temperatures between 200-300℃, the ultimate load of the FRP composites significantly increased from 731.01N to 1650.97N and increased 133.51% of the initial ultimate load, which indicates that the FRP composites had experienced a strengthening process. The reason for this phenomenon is that the matrix bonded by the fiber resin changes with the increase in temperature, when exposed to temperatures below 200℃, the mechanical properties slightly increase, especially when the exposure temperatures are between 200-300℃, this could be attributed to that the bonding effects of nanometer montmorillonite and fiber material are most obvious at those temperatures, which lead to the increment of ultimate load. Notably, the ultimate load of FRP composites significantly decreased from 1650.97N to 252.24N and reduced 64.32% of the initial ultimate load when exposed to temperatures of 350℃, this could be attributed to the bonding of nanometer montmorillonite and fiber material is softened at this temperature and the resin matrix enters the rubber state from the glass state, in which the transition temperature Tg is nearly 300℃ based on the test results.

Fracture load: However, the maximum increase in the fracture load of the FRP composites was 541.35N and 963.26% of their initial fracture load when exposed to temperatures between 200-300℃, which is consistent with the phenomenon of ultimate load, this could be attributed to that the bonding effects of nanometer montmorillonite and fiber material are most obvious at those temperatures, which lead to the increment of fracture load. Furthermore, the fracture load of the FRP composites significantly decreased from 597.55N to 133.62N when exposed to temperatures of 350℃, this could be attributed to the bonding of nanometer montmorillonite and fiber material is softened at this temperature and the resin matrix enters the rubber state from the glass state, in which the transition temperature Tg is nearly 300℃ based on the test results.

Fracture displacement: Particularly when the FRP composites were exposed to 300℃, the maximum increase in fracture displacement was 1.56mm and 87% of their initial fracture displacement, this could be attributed to that the bonding effects of nanometer montmorillonite and fiber material are most obvious at those temperatures, which lead to the mass increment of ductility, causing the fracture displacement to increase. It is worth noting that when ex-posed to temperatures above 300℃, the fracture displacement of the FRP composites de-creased from 3.37mm to 2.46mm, this could be attributed to the bonding of nanometer montmorillonite and fiber material is softened at this temperature and the resin matrix enters the rubber state from the glass state, in which the transition temperature Tg is nearly 300℃ based on the test results.

Elastic modulus: However, when exposed to temperatures between 200-300℃, the elastic modulus of the FRP composites significantly increased from 820.32MPa to 1181.44MPa and increased 47.39% of the initial elastic modulus, which indicated that the FRP composites had expe-rienced a strengthening process, this could be attributed to that the bonding effects of nanometer montmorillonite and fiber material are most obvious at those temperatures, which lead to the increment of elastic modulus. Notably, the elastic modulus of FRP composites significantly decreased from 1181.44MPa to 752.75MPa and reduced by 6.1% of the initial elastic modulus when exposed to temperatures of 350℃, this could be attributed to the bonding of nanometer montmorillonite and fiber material is softened at this temperature and the resin matrix enters the rubber state from the glass state, in which the transition temperature Tg of the resin matrix is nearly 300℃ based on the test results.

Thanks again for your invaluable comments. These useful comments improved not only the quality of our manuscript, but also our writing skills.

Reviewer 2 Report

Previous comments addressed

Author Response

Thanks again for your invaluable comments. These useful comments improved not only the quality of our manuscript, but also our writing skills.

Reviewer 3 Report

Analyzing the authors' answers, I found that the authors did not respond concretely to all the mentioned comments. In addition, some additions were requested both in terms of comparison with the literature and in terms of improving certain sections, and the authors not taking these tips into account. Therefore, following these findings, the decision remains the same with the mention of responding promptly and concisely to previously unresolved comments.

1.      In the response to reviewer remark 1, the authors say that are mentioning natural fibres to “introduce fibre-reinforced polymer composite”. Please note that the basalt fibres you are using are not natural, so the information about natural fibres are completely out of context. Instead, you should mention mineral or synthetic fibres.

2.      Concerning the additional information about the effect of elevated temperature on FRP materials; again, the authors put information which are not consistent with the topic addressed by the paper. In particular, the authors reported information about the influence of elevated temperature on the mechanical behaviour of concrete elements reinforced with FRP. Please note that in your study you are evaluating the mechanical behaviour of FRP as a standalone material. You should refer to work related to the material characterization tests of FRP at elevated temperatures. Please check the suggestions reported in the previous round.

3.      Please don’t say that the information available in the literature about the influence of elevated temperature on FRP materials are scarce, this is not true; there are plenty of information available. Instead, you should highlight how your work improves the current state of the art.

4.      In the response to reviewer remark 2, the authors simply summarized the experimental campaign carried out without emphasizing the aim of the study. Please correct.

5.      In the response to reviewer remark 3, it is strongly suggested to improve the English writing. What you wrote is very difficult to read. In addition, in contrast with the authors reply, no information is provided about the thermos-mechanical properties of the FRP material.

6.      In the response to reviewer remarks 6 and 7, the authors failed to clarify the doubts raised by the reviewer. First, it is not clear if the temperature considered in this study corresponds to the temperature of the air in the thermal chamber or within the specimen. Then, please note that your procedure to compute axial strain is wrong. It is not possible to compute axial strain by using the cross-head displacement of the machine. This is not scientifically correct!

7.      In the response to reviewer remark 7, the authors’ reply is very weak. The reason for having that non-monotonic variation trend is not clear; from what you wrote, progressive reductions of the mechanical properties should be expected.

Author Response

We appreciate very much the time and effort that the reviewer spent on this paper. We have carefully read and considered the comments made by the reviewer, and these comments are valuable to improve the quality manuscript. According to these comments, we conducted a detailed revision. All the comments are addressed, and the revised contents are printed in red in the text. All changes concerning the comments or answers to the questions are shown below.

Reviewer: #1: Analyzing the authors' answers, I found that the authors did not respond concretely to all the mentioned comments. In addition, some additions were requested both in terms of comparison with the literature and in terms of improving certain sections, and the authors not taking these tips into account. Therefore, following these findings, the decision remains the same with the mention of responding promptly and concisely to previously unresolved comments.

Thank you a lot for your invaluable comments. According to your suggestions, we have modified all the mentioned comments again.

  1. In the response to reviewer remark 1, the authors say that are mentioning natural fibers to “introduce fibre-reinforced polymer composite”. Please note that the basalt fibres you are using are not natural, so the information about natural fibres are completely out of context. Instead, you should mention mineral or synthetic fibres.

Reply: Thanks for your helpful suggestion. In the reviewed manuscript, the information about natural fibres has been completely out of context. As you suggested, we have deleted this part so as not to distract the readers from the main topic of this paper. And we have reorganized the introduction of this reviewed manuscript. The modifications are as follows:

The deleted parts are as follows:

Due to problems like global warming and rising demand from environmental activists in emerging countries to create better degradable materials, natural composites are preferred in current products. Reinforced fiber composites are manufactured from plant fibers and utilized in a variety of high-profile applications, including helmets, architectural designs, and vehicle spare parts [1, 2]. Generally speaking, natural fiber is extracted from plants, animals, and minerals. Natural materials like biomass resources fibers are utilized more frequently since they better attach to and link with the resin they are reinforced with [3].

Every composite made with natural fibers has this fundamental drawback: it is extremely combustible. Energy-producing resources are vulnerable to heat and fire in common applications like aerospace and automotive [4]. Flame retardant material is utilized as an addition in a specific ratio to overcome the composite material's flammability and lessen the candlewick effect. Strong adhesion between the fiber and matrix is achieved, increasing the material's capacity to carry larger loads [5]. The most prevalent thermal insulator is fire retardant. The composite's thermal conductivity decreases as the quantity of fiber material increases. Even slightly enhancing the fire retardant percentage inevitably has a significant impact on the strength of the material [6]. The inorganic fire-resistant compounds are more hygroscopic than untreated wood at high humidity levels [7]. The most recommended treatment for all-natural fibers is alkali treatment as it rapidly eliminates the organic components (lignin, cellulose), which have a significant impact on the mechanical and thermal characteristics of FRP composite material. Tensile strength is increased by 10% to 20%. Meanwhile, excessive treatment weakens the connections between fibers, reducing the strength [8].

After being exposed to 120, 130, 150, and 180 °C, respectively, GFRP-strengthened concrete cylinders' ultimate axial compressive strength was reduced by 2%, 4%, 13%, and 18% [11]. However, at 150 and 185 °C, the ultimate compressive strength of GFRP-strengthened concrete cylinders with an epoxy-based fireproofing coating reduced by around 3% and 10%. At various temperatures, the failure mechanisms of GFRP-strengthened concrete cylinders have been identified as fiber-dominated at lower temperatures and resin-dominated at higher temperatures. Chowdhury [12] has studied how full-scale reinforced concrete cylinders (400 3810 mm) with exterior fire insulation that contains an FRP-reinforced layer alter in structural qualities when exposed to fire. The interior reinforced bar and concrete of FRP-wrapped reinforced concrete buildings maintained a lower temperature for up to 300 minutes due to the exterior fire insulation. Nevertheless, during 34 minutes, the temperature of the FRP-reinforced layer insulated by 53 mm of fire insulation stayed below its Tg.

The added parts are as follows:

The above scholars mainly focused on the mechanical behavior of concrete elements reinforced with FRP. There are studies about the mechanical behavior of FRP as a standalone material at elevated temperatures. Pultruded carbon fiber reinforced polymer (P-CFRP) specimens and CFRP tensile specimens manufactured with a hand lay-up method were subjected to a series of tests by Nguyen et al. [3,4] at temperatures that reached 700 °C. According to their findings, hand layup specimens' ultimate tensile strength and Young's modulus were reduced by 50% at 350 °C and 30% at 600 °C, respectively. Additionally, they demonstrated that the thermomechanical strength is lower than the residual strength for P-CFRP samples at the same degree of applied temperature. One of the pioneering studies regarding the behavior and characteristics of FRP materials at high temperatures that are utilized in industrial domains, such as the automotive, marine, and aerospace industries, was performed by Mouritz and Mathys [5]. At high temperatures, Shenghu and Zhishen [6] performed a series of tension tests on single-layer FRP sheets composed of GFRP, CFRP, and basalt-fiber reinforced (BFRP). Among all the tested fiber-reinforced sheets, they concluded that the CFRP sheets had the highest strength and the GFRP sheets had the lowest strength [7]. At around 55°C, all of the sheets' tensile strength significantly decreased, but no further substantial decline occurred as the temperature increased. The CFRP sheets had the highest residual strength, with almost 69% of their initial tensile strength. However, there still lacks the work of establishing the constitutive model to better predict the mechanical behavior of FRPs at elevated temperatures. In this research, we proposed a constitutive model based on the experimental results of FRPs at elevated temperatures to fill the research gap.

  1. Concerning the additional information about the effect of elevated temperature on FRP materials; again, the authors put information which is not consistent with the topic addressed by the paper. In particular, the authors reported information about the influence of elevated temperature on the mechanical behaviour of concrete elements reinforced with FRP. Please note that in your study you are evaluating the mechanical behaviour of FRP as a standalone material. You should refer to work related to the material characterization tests of FRP at elevated temperatures. Please check the suggestions reported in the previous round.

Reply: Thanks for your helpful suggestion. Indeed, as you said, in the original manuscript, we have reported much information about the influence of elevated temperature on the mechanical behaviour of concrete elements reinforced with FRP. According to your suggestion, we have added the research information about the mechanical properties of FRP at elevated temperatures, especially the mechanical behavior of FRP as a standalone material at high temperatures. The modifications are as follows:

The above scholars mainly focused on the mechanical behavior of concrete elements reinforced with FRP. There are studies about the mechanical behavior of FRP as a standalone material at elevated temperatures. Pultruded carbon fiber reinforced polymer (P-CFRP) specimens and CFRP tensile specimens manufactured with a hand lay-up method were subjected to a series of tests by Nguyen et al. [3,4] at temperatures that reached 700 °C. According to their findings, hand lay-up specimens' ultimate tensile strength and Young's modulus were reduced by 50% at 350 °C and 30% at 600 °C, respectively. Additionally, they demonstrated that the thermomechanical strength is lower than the residual strength for P-CFRP samples at the same degree of applied temperature. One of the pioneering studies regarding the behavior and characteristics of FRP materials at high temperatures that are utilized in industrial domains, such as the automotive, marine, and aerospace industries, was performed by Mouritz and Mathys [5]. At high temperatures, Shenghu and Zhishen [6] performed a series of tension tests on single-layer FRP sheets composed of GFRP, CFRP, and basalt-fiber reinforced (BFRP). Among all the tested fiber-reinforced sheets, they concluded that the CFRP sheets had the highest strength and the GFRP sheets had the lowest strength [7]. At around 55°C, all of the sheets' tensile strength significantly decreased, but no further substantial decline occurred as the temperature increased. The CFRP sheets had the highest residual strength, with almost 69% of their initial tensile strength.

In this research, we conducted an experimental study of FRP composite as a standalone material at high temperatures ranging from 20 °C to 350 °C and proposed a constitutive JC model which considers the impact of elevated temperatures.

  1. Please don’t say that the information available in the literature about the influence of elevated temperature on FRP materials are scarce, this is not true; there is plenty of information available. Instead, you should highlight how your work improves the current state of the art.

Reply: Thanks for your helpful suggestion. Regarding the research work on high-temperature mechanical properties of FRP we do express the inaccuracies in the original manuscript. Some studies have provided experimental results on the mechanical behaviour of FRP at elevated temperatures. However, most of the tested specimens are FRP bars and a few are FRP plates. Meanwhile, the existing research results have found that the composition of the FRP material, the type of bonding colloid, and the manufacturing process have different effects on the high-temperature mechanical behaviour of FRP materials. According to your suggestion, we have added the research works about the influence of elevated temperature on FRP materials and have highlighted how the work improves the current state of the art. The modifications are as follows:

There are studies about the mechanical behavior of FRP as a standalone material at elevated temperatures. Pultruded carbon fiber reinforced polymer (P-CFRP) specimens and CFRP tensile specimens manufactured with a hand lay-up method were subjected to a series of tests by Nguyen et al. [3,4] at temperatures that reached 700 °C. According to their findings, hand lay-up specimens' ultimate tensile strength and Young's modulus were reduced by 50% at 350 °C and 30% at 600 °C, respectively. Additionally, they demonstrated that the thermomechanical strength is lower than the residual strength for P-CFRP samples at the same degree of applied temperature. One of the pioneering studies regarding the behavior and characteristics of FRP materials at high temperatures that are utilized in industrial domains, such as the automotive, marine, and aerospace industries, was performed by Mouritz and Mathys [5]. At high temperatures, Shenghu and Zhishen [6] performed a series of tension tests on single-layer FRP sheets composed of GFRP, CFRP, and basalt-fiber reinforced (BFRP). Among all the tested fiber-reinforced sheets, they concluded that the CFRP sheets had the highest strength and the GFRP sheets had the lowest strength [7]. At around 55°C, all of the sheets' tensile strength significantly decreased, but no further substantial decline occurred as the temperature increased. The CFRP sheets had the highest residual strength, with almost 69% of their initial tensile strength. However, there still lacks the work of establishing the constitutive model to better predict the mechanical behavior of FRPs at elevated temperatures. In this research, we proposed a constitutive model based on the experimental results of FRPs at elevated temperatures to fill the research gap.

  1. In the response to reviewer remark 2, the authors simply summarized the experimental campaign carried out without emphasizing the aim of the study.

Reply: Thanks for your helpful suggestion. According to your suggestion, we have added the aim of this study at the end of the introduction section. The modifications are as follows:

In recent years, nanometer montmorillonite composite fiber material has gradually been used in building structures. As a new fiber-reinforced polymer (FRP) composite, the change in mechanical properties of nanometer montmorillonite composite fiber-reinforced bars or plates has not been investigated. Therefore, the mechanical behavior of nanometer montmorillonite composite fiber-reinforced plates subjected to different temperatures was studied. The experimental results provided in this paper can be applied to both further research and engineering applications when conducting theoretical analysis and numerical simulation of nanometer montmorillonite composite fiber-reinforced polymer (FRP) composites. And this research is a part of the larger experimental program that aims to examine the mechanical characteristics and behavior of nanometer montmorillonite composite fiber-reinforced polymer (FRP) composites in various situations.

  1. In the response to reviewer remark 3, it is strongly suggested to improve the English writing. What you wrote is very difficult to read. In addition, in contrast with the authors reply, no information is provided about the thermos-mechanical properties of the FRP material.

According to your suggestion, the writing of this paper has been carefully checked and revised. The test specimen in this research is provided by China Southwest Architecture Design and Research Institute and we have obtained the material characteristics of FRPs recently. The thermos-mechanical properties of the FRP material including ultimate load, fracture load, fracture displacement and elastic modulus are discussed in the next part of the paper. The following are its detailed material characteristics at ambient temperature:

In this test, physical and mechanical properties of nanometer montmorillonite com-posite fiber material, including the density ρ, Barcol hardness, fiber volume fraction, in-soluble content of resin, water absorption, glass transition temperature Tg, tensile strength (main fiber direction) ƒtm, tensile strength (secondary fiber direction) ƒts, compressive strength (main fiber direction) ƒcm, compressive strength (secondary fiber direction) ƒcs and shock resistance of nanometer montmorillonite composite fiber material are provided in Table 1.

Table 1 Physical and mechanical properties of nanometer montmorillonite composite fiber material.

Performance

Performance index

ρ / (kg•m-3)

≤ 2000

Barcol hardness / (HBa)

≥ 50

Fiber volume fraction / %

≥ 70

Insoluble content of resin / %

≥ 90

Water absorption / %

≤ 1.0

Tg / ℃

≥ 290

ƒtm / MPa

≥ 400

ƒts / MPa

≥ 10

ƒcm / MPa

≥ 100

ƒcs / MPa

≥ 15

Shock resistance / (kJ•m-2)

≥ 240

  1. In the response to reviewer remarks 6 and 7, the authors failed to clarify the doubts raised by the reviewer. First, it is not clear if the temperature considered in this study corresponds to the temperature of the air in the thermal chamber or within the specimen. Then, please note that your procedure to compute axial strain is wrong. It is not possible to compute axial strain by using the cross-head displacement of the machine. This is not scientifically correct!

Reply: Thanks for your helpful suggestion. The testing machine is equipped with high temperature test chamber device (as shown in Fig.2), which can test the tensile, compression, bending and shearing mechanical properties of metal and non-metal materials under high temperature environment, and meet the test standards such as GB/ISO/ASTM/JIS/DIN. The test machine connected to the computer through the microcomputer control system can automatically obtain the maximum strength, maximum deformation, stress and strain test data. First, the target temperature is controlled by the thermocouples in the test equipment, whether heating to a specified temperature or keeping the temperature constant. In this test, we consider ambient temperature、50℃、100℃、150℃、200℃、250℃、300℃ and 350℃as test conditions and compare those target temperatures in the next part of the paper, and the ambient temperature is 20℃in this test. Then, in this study, we adopted the displacement control instead of the strain control due to my inappropriate description in the paper, in which the loading method conformed to the requirements of GB/T 228.1-2010. Due to the test equipment, we could not simultaneously test the specimen and attach the strain gauge at high temperatures. However, at the beginning of the test, we input the information of the specimen into the test equipment, when the specimen is fractured, the test equipment can output both the displacement and engineering strain of the specimen, then we compare the strain output by the computer with the strain obtained by the traditional method , where L is the gauge length after deformation and L 0 is the gauge length and L- L 0 can be reflected by the displacement, the two results are basically the same, as shown in Fig.3. Therefore, considering the effect of thermal strain, we only discuss the load-displacement curve rather than the stress-strain curve in the test results, while we use the traditional calculation formula to calculate the strain when establishing the constitutive model. Some descriptions are as follows:

To ensure uniform temperature over the entire gauge length, the specimen was kept for about 30 min at each target temperature and the elevated temperatures remained unchanged based on the thermocouples in the test equipment. Then the specimen was loaded until it fails, during which the target temperatures were unchanged since the specimen was still in the test equipment and the tensile displacement rate was constant and the loading increased as the temperature increased. Both the displacement and engineering strain of the FRPs can be output by the computer.

Figure 2. Test machine

Figure 3. Comparison between strain obtained by test equipment and formula

  1. In the response to reviewer remark 7, the authors’ reply is very weak. The reason for having that non-monotonic variation trend is not clear; from what you wrote, progressive reductions of the mechanical properties should be expected.

Reply: Thank you a lot for your invaluable comments. Indeed, as you said, in the original manuscript, we reported less information about the reason for explaining the non-monotonic variation trend. According to your suggestions, we have modified the descriptions about the mechanism of how the elevated temperature affects the mechanical properties based on the test results. Besides, all the test procedures are carried out following the specifications, and the test results are accurate. But the exact reason that can accurately account for the change is unclear, we are discussing the results of the test. Some descriptions are added in the paper to clarify this issue:

Ultimate load: However, when exposed to temperatures between 200-300℃, the ultimate load of the FRP composites significantly increased from 731.01N to 1650.97N and increased 133.51% of the initial ultimate load, which indicates that the FRP composites had experienced a strengthening process. The reason for this phenomenon is that the matrix bonded by the fiber resin changes with the increase in temperature, when exposed to temperatures below 200℃, the mechanical properties slightly increase, especially when the exposure temperatures are between 200-300℃, this could be attributed to that the bonding effects of nanometer montmorillonite and fiber material are most obvious at those temperatures, which lead to the increment of ultimate load. Notably, the ultimate load of FRP composites significantly decreased from 1650.97N to 252.24N and reduced 64.32% of the initial ultimate load when exposed to temperatures of 350℃, this could be attributed to the bonding of nanometer montmorillonite and fiber material is softened at this temperature and the resin matrix enters the rubber state from the glass state, in which the transition temperature Tg is nearly 300℃ based on the test results.

Fracture load: However, the maximum increase in the fracture load of the FRP composites was 541.35N and 963.26% of their initial fracture load when exposed to temperatures between 200-300℃, which is consistent with the phenomenon of ultimate load, this could be attributed to that the bonding effects of nanometer montmorillonite and fiber material are most obvious at those temperatures, which lead to the increment of fracture load. Furthermore, the fracture load of the FRP composites significantly decreased from 597.55N to 133.62N when exposed to temperatures of 350℃, this could be attributed to the bonding of nanometer montmorillonite and fiber material is softened at this temperature and the resin matrix enters the rubber state from the glass state, in which the transition temperature Tg is nearly 300℃ based on the test results.

Fracture displacement: Particularly when the FRP composites were exposed to 300℃, the maximum increase in fracture displacement was 1.56mm and 87% of their initial fracture displacement, this could be attributed to that the bonding effects of nanometer montmorillonite and fiber material are most obvious at those temperatures, which lead to the mass increment of ductility, causing the fracture displacement to increase. It is worth noting that when ex-posed to temperatures above 300℃, the fracture displacement of the FRP composites de-creased from 3.37mm to 2.46mm, this could be attributed to the bonding of nanometer montmorillonite and fiber material is softened at this temperature and the resin matrix enters the rubber state from the glass state, in which the transition temperature Tg is nearly 300℃ based on the test results.

Elastic modulus: However, when exposed to temperatures between 200-300℃, the elastic modulus of the FRP composites significantly increased from 820.32MPa to 1181.44MPa and increased 47.39% of the initial elastic modulus, which indicated that the FRP composites had expe-rienced a strengthening process, this could be attributed to that the bonding effects of nanometer montmorillonite and fiber material are most obvious at those temperatures, which lead to the increment of elastic modulus. Notably, the elastic modulus of FRP composites significantly decreased from 1181.44MPa to 752.75MPa and reduced by 6.1% of the initial elastic modulus when exposed to temperatures of 350℃, this could be attributed to the bonding of nanometer montmorillonite and fiber material is softened at this temperature and the resin matrix enters the rubber state from the glass state, in which the transition temperature Tg of the resin matrix is nearly 300℃ based on the test results.

Round 3

Reviewer 3 Report

Although the authors made some revisions, most of the comments were not reflected in the revised paper. Some key responses did not directly reflect to the reviewers' comments. Some basic properties analysis and mechanism summaries of GFRP are still missing. The is plenty of information highlighted in previous revision rounds which is very cursory and does not correspond to the truth but most important I have serious doubts (raised in the previous rounds) about the methodologies used in the experimental campaign. In addition, the contribution and innovation of current research work are not clear. 

Based on this, the reviewer deems it more appropriate to continue indicating a major revision of the manuscript but refers to the editor's evaluation regarding the decision to accept the work for publication in the Journal.

Author Response

Response to Comments from Reviewer No. 1:

We appreciate very much the time and effort that the reviewer spent on this paper. We have carefully read and considered the comments made by the reviewer, and these comments are valuable to improve the quality manuscript. According to these comments, we conducted a detailed revision. All the comments are addressed, and the revised contents are printed in red in the text. All changes concerning the comments or answers to the questions are shown below.

Reviewer: #1: Although the authors made some revisions, most of the comments were not reflected in the revised paper. Some key responses did not directly reflect to the reviewers' comments. Some basic properties analysis and mechanism summaries of GFRP are still missing. The is plenty of information highlighted in previous revision rounds which is very cursory and does not correspond to the truth but most important I have serious doubts (raised in the previous rounds) about the methodologies used in the experimental campaign. In addition, the contribution and innovation of current research work are not clear. Based on this, the reviewer deems it more appropriate to continue indicating a major revision of the manuscript but refers to the editor's evaluation regarding the decision to accept the work for publication in the Journal.

Thank you a lot for your invaluable comments. According to your suggestions, some revised descriptions are added to the paper, while some descriptions just aim to explain your confusion, which is not suitable to reflect in the revised paper. All the comments were answered in the last response point by point. Indeed, as you said, in the original manuscript, we discussed some basic properties analysis and mechanism summaries of GFRP. However, other scholars study the mechanical properties of FRP materials (CFRP and GFRP) at high temperatures, but also study its mechanical properties, which includs elastic modulus, fracture strength, fracture displacement and ultimate strength at high temperatures. In this paper, we also investigate the four crucial mechanical properties of nanometer montmorillonite composite fiber material at elevated temperatures. And according to your suggestions, the mechanism summaries of nanometer montmorillonite composite fiber material are also added in the manuscript to better understand the changes in mechanical properties. Besides, we have submitted the specific material parameters you asked us to provide in the last revision, including the conversion temperature Tg, as shown in Table 1.

Table 1 Physical and mechanical properties of nanometer montmorillonite composite fiber material.

Performance

Performance index

ρ / (kg•m-3)

≤ 2000

Barcol hardness / (HBa)

≥ 50

Fiber volume fraction / %

≥ 70

Insoluble content of resin / %

≥ 90

Water absorption / %

≤ 1.0

Tg / ℃

≥ 290

ƒtm / MPa

≥ 400

ƒts / MPa

≥ 10

ƒcm / MPa

≥ 100

ƒcs / MPa

≥ 15

Shock resistance / (kJ•m-2)

≥ 240

    In terms of the test process, we have also made a supplementary explanation to your doubts raised last time, including the parameters of the test equipment and the method of strain calculation, in which the results are obtained by comparing experimental data and test equipment. The added description is that considering the effect of thermal strain, we only discuss the load-displacement curve rather than the stress-strain curve in the test results, while we use the traditional calculation formula to calculate the strain when establishing the constitutive model. And the two results are basically the same, as shown in Fig.1.

Fig.1 Comparison between strain obtained by test equipment and formula

In addition, the contribution and innovation of current research are added in the last part of the introduction, which can be expressed as: The experimental results provided in this paper can be applied to both further research and engineering applications when conducting theoretical analysis and numerical simulation of nanometer montmorillonite composite fiber-reinforced polymer (FRP) composites. And this research is a part of the larger experimental program that aims to examine the mechanical characteristics and behavior of nanometer montmorillonite composite fi-ber-reinforced polymer (FRP) composites in various situations.

Finally, the writing of this paper has been carefully checked and revised again.

Round 4

Reviewer 3 Report

Based on the additional comments and insights provided by the authors, the Reviewer states a positive opinion for the publication of the manuscript.

The new comments help the reader better understand the nature and scope of the research, although some points still remain unanswered.

Finally, by providing the publication opportunity, the Reviewer wants to reward the effort put into the review rounds and hopes that the Authors will use the suggestions constructively in the future.